# ADVERSARIAL CHEAP TALK

## ABSTRACT

Adversarial attacks in reinforcement learning (RL) often assume highly-privileged access to the victim's parameters, environment, or data. Instead, this paper proposes a novel adversarial setting called a *Cheap Talk MDP* in which an Adversary can merely append deterministic messages to the Victim's observation, resulting in a *minimal range of influence*. The Adversary cannot occlude ground truth, influence underlying environment dynamics or reward signals, introduce non-stationarity, add stochasticity, see the Victim's actions, or access their parameters. Additionally, we present a simple meta-learning algorithm called Adversarial Cheap Talk (ACT) to train Adversaries in this setting. We demonstrate that an Adversary trained with ACT can still significantly influence the Victim's *training and testing* performance, despite the highly constrained setting. Affecting train-time performance reveals a new attack vector and provides insight into the success and failure modes of existing RL algorithms. More specifically, we show that an ACT Adversary is capable of *harming* performance by interfering with the learner's function approximation, or instead *helping* the Victim's performance by outputting useful features. Finally, we show that an ACT Adversary can manipulate messages during train-time to directly and arbitrarily control the Victim at test-time.

## 1 INTRODUCTION

Learning agents are often trained in settings where adversaries are not able to influence underlying environment dynamics or reward signals, but may influence part of the agent's observations. For instance, adversaries may append arbitrary tags to content that will be used to train recommender systems. Similarly, they may rent space on interactive billboards near busy traffic intersections to influence data sets used for training self-driving cars. In financial markets, adversaries can alter the state of the order-book by submitting orders far out of the money (essentially for free). While these features are 'useless' from an information-theoretic point of view, it is common practice in end-to-end deep learning to include them as part of the input and let the model learn which features matter. For instance, self-driving cars typically do not omit useless parts of the visual input but instead learns to ignore them through training. Surprisingly, this paper demonstrates that an actor can still heavily influence the behaviour and performance of learning agents by controlling information *only* in these 'useless' channels, *without* knowing anything about the agent's parameters or training state.

Most past work in adversarial RL assumes that adversaries can influence environment dynamics (Huang et al., 2017; Gleave et al., 2020). For example, perturbing images and observations could obscure or alter relevant information, such as the ball's location in a Pong game (Kos and Song, 2017). Furthermore, many attacks require access to the trained agent's weights and parameters to generate the adversarial inputs (Wang et al., 2021). Finally, most of these attacks only cause the victim's policy to fail arbitrarily instead of giving the adversary full control over the victim's policy at test time (Gu et al., 2017; Kiourti et al., 2020; Salem et al., 2020; Ashcraft and Karra, 2021; Zhang et al., 2021).

Instead, we propose a novel minimum-viable setting called *Cheap Talk MDP*s. Adversaries are only allowed to modify 'useless' features that are appended to the Victim's observation as a deterministic function of the current state. These features represent parts of an agent's observations that are unrelated to rewards or transition dynamics. In particular, our model applies to Adversaries adding tags to content in recommender systems, or renting space on interactive billboards, or submitting orders far out of the money in financial markets. The setting is *minimal* in that Adversaries cannot use these features to occlude ground truth, influence environment dynamics or reward signals, inject stochasticity, introduce non-stationarity, see the Victim's actions, or access their parameters.

Cheap Talk MDPs are formalised in Section 4, and we further justify minimality by proving in Proposition 1 that Adversaries cannot influence *tabular* Victims *whatsoever* in Cheap Talk MDPs. In particular, it follows that Adversaries can only influence Victims by interfering with their function approximator. We also prove more generally in Proposition 2 that Adversaries cannot prevent Victims with optimal convergence guarantees to converge to optimal rewards, even in non-tabular settings.

Despite these restrictions, we show that Adversaries can still heavily influence agents parameterised by neural networks. In Section 5, we introduce a new meta-learning algorithm called Adversarial Cheap Talk (ACT) to train the Adversary. With an extensive set of experiments, we demonstrate in Section 6 that an ACT Adversary can manipulate a Victim to achieve a number of outcomes:

1. An ACT Adversary can prevent the Victim from solving a task, resulting in low rewards *during training*. We provide empirical evidence that the Adversary successfully sends messages which induce *catastrophic interference* in the Victim's neural network.

2. Conversely, an ACT Adversary can learn to send useful messages that *improve* the Victim's training process, resulting in higher rewards *during training*.

3. Finally, we introduce a training scheme that allows the ACT Adversary to directly and arbitrarily control the Victim directly *at test-time*.

## 2 RELATED WORK

### 2.1 TEST-TIME ADVERSARIAL ATTACKS

Most work investigating adversarial attacks on deep RL systems focuses on attacks at test-time, i.e. those that attack a fully trained, static policy. Huang et al. (2017), Kos and Song (2017), and Zhang et al. (2021) investigate adversarial attacks that influence test-time performance by directly perturbing the observation space. Unlike in Cheap Talk MDPs, such perturbations can influence the underlying dynamics by obscuring relevant information. Gleave et al. (2020) investigate adversarial attacks that influence test-time performance of reinforcement learning agents that were trained in self-play. In contrast to our method, the adversarial agent can directly interact with the environment and the victim agent. The aforementioned test-time attacks largely work by generating perturbations that push the observations out of the Victim's training distribution. In contrast, in Cheap Talk MDPs, the Victim *trains directly with the static adversarial features*; thus, by definition, the Adversary cannot generate out-of-distribution or non-stationary inputs. Furthermore, these test-time attacks also assume the ability to directly train against a pre-trained static agent. In contrast, in Cheap Talk MDPs, the adversary is influences a learning agent with random initial parameters.

### 2.2 TRAIN-TIME ADVERSARIAL ATTACKS

In contrast to train-time adversarial attacks in RL, in test-time adversarial attacks the dversary interacts with a *learning* victim. Pinto et al. (2017) simultaneously trains an adversary alongside a reinforcement learning agent to robustify the victim's policy. Unlike in this work, the adversary is able to directly apply perturbation forces to the environment. We make further comparisons in Section 6.1. Backdoor attacks in reinforcement learning aim to introduce a vulnerability during train-time, which can be triggered at test-time. Kiourti et al. (2020) and Ashcraft and Karra (2021) assume the adversary can directly and fully modify the victim's observations and rewards in order to discretely insert a backdoor that triggers on certain inputs. This is unlike Cheap Talk MDPs, in which only 'useless' parts of the observations can be modified. Wang et al. (2021) considers the multi-agent setting where the adversary inserts a backdoor using its behaviour in the environment. Unlike in this work, the adversary can influence the underlying environment dynamics. Furthermore, each of these backdoor attacks simply cause the victim to fail when triggered. In contrast, we use the backdoor to fully control the victim.

### 2.3 FAILURE MODES IN DEEP REINFORCEMENT LEARNING

Previous works have shown that using neural networks as function approximators in reinforcement learning often results in multiple failure modes due to the non-stationarity of value function bootstrapping (van Hasselt et al., 2018). In particular, works have shown that catastrophic interference

(Bengio et al., 2020) and capacity loss (Lyle et al., 2022) often occur, even within a single episode of an environment (Fedus et al., 2020). Song et al. (2020) shows that deep reinforcement learning algorithms can often overfit to spurious correlations in the observation space. By appending to the observation space, we learn to induce the observational failure modes described in these works.

## 2.4 OPPONENT SHAPING / CHEAP TALK

Our method is closely related to the field of opponent shaping. Originally, most opponent shaping algorithms assumed white-box access to their opponents to shape the flow of the opponent's gradient (Foerster et al., 2018; Letcher et al., 2019a;b; Willi et al., 2022). Instead, Lu et al. (2022) introduce a method to shape opponents without white-box access. However, they still deploy an agent to interact directly in the environment. In contrast, we propose a method to shape other agents without having to interact in the environment, solely by appending messages through a cheap talk channel. In reinforcement learning, a cheap talk channel is a part of the state space which can be observed by other agents but does not alter transition dynamics or reward functions. Cheap talk channels (Crawford and Sobel, 1982) in deep reinforcement learning have been used to learn emergent communication (Foerster et al., 2016) and to solve coordination problems (Cao et al., 2018). To the best of our knowledge, this paper is the first to use a cheap talk channel (and only a cheap talk channel) to shape the behaviour of learning agents.

# 3 BACKGROUND

## 3.1 REINFORCEMENT LEARNING

A Markov decision process (MDP) consists of a tuple $\mathcal{D} = \langle \mathcal{S}, \mathcal{A}, \mathcal{P}, \mathcal{R}, \gamma \rangle$, where $\mathcal{S}$ denotes the state space, $\mathcal{A}$ represents the action space, $\mathcal{P} : \mathcal{S} \times \mathcal{A} \times \mathcal{S} \mapsto [0, 1]$ denotes the state transition probability function, $\mathcal{R} : \mathcal{S} \times \mathcal{A} \mapsto \mathbb{R}$ is the reward function and $\gamma \in [0, 1)$ denotes the discount factor. At every timestep $t$, an agent samples an action from its stochastic policy $a_t \sim \pi_\theta (\cdot \mid s_t)$, where $a_t \in \mathcal{A}$, $s_t \in \mathcal{S}$ and $\theta$ denotes the policy parameterisation. The agent then receives a reward based on the action taken in the current state: $r_t = \mathcal{R}(s_t, a_t)$. Finally, a new state is sampled according to the transition function $s_{t+1} \sim \mathcal{P}(\cdot \mid s_t, a_t)$, resulting in a trajectory $\tau_\theta := ((s_0, a_0, r_0), (s_1, a_1, r_1), \ldots)$. The agent's goal is to maximise its expected discounted return under policy $\pi_\theta$:

$$J(\theta) = \mathbb{E}_{\pi_\theta} \left[ \sum_{t=0}^{\infty} \gamma^t r_t \right] . \tag{1}$$

## 3.2 EVOLUTION STRATEGIES

Evolution Strategies (Salimans et al., 2017, ES) is a derivative-free optimisation method. Let $F : \mathbb{R}^d \to \mathbb{R}$ be some function we want to optimise over. Instead of optimising $F(\mathbf{x})$ directly, we first blur the objective to

$$\mathbb{E}_{\epsilon \sim N(\mathbf{0}, I_d)}[F(\mathbf{x} + \sigma \epsilon)] ,$$

where $\sigma$ is a hyper-parameter dictating the amount of Gaussian noise we add. This is useful because

$$\nabla_{\mathbf{x}} \mathbb{E}_{\epsilon \sim N(\mathbf{0}, I_d)}[F(\mathbf{x} + \sigma \epsilon)] = \mathbb{E}_{\epsilon \sim N(\mathbf{0}, I_d)} \left[ \frac{\epsilon}{\sigma} F(\mathbf{x} + \sigma \epsilon) \right] ,$$

which allows us to optimise a non-differentiable function using gradient descent techniques in a highly scalable manner.

# 4 PROBLEM SETTING

The setting we propose is of two agents interacting in a *Cheap Talk MDP* $\langle \mathcal{S}, \mathcal{A}, \mathcal{P}, \mathcal{R}, \gamma, \mathcal{M}, f, \mathcal{J} \rangle$, which is effectively an MDP with an augmented state space, whereby features (messages) from a Cheap Talk channel $\mathcal{M}$ are appended to states. We refer to the agent observing the augmented state as the *Victim*, with transition and reward functions $\mathcal{P}, \mathcal{R}$ assumed to be independent from $\mathcal{M}$. Formally, this means that $\mathcal{P}(\cdot \mid s, m, a) = \mathcal{P}(\cdot \mid s, m', a)$ and $\mathcal{R}(s, m, a) = \mathcal{R}(s, m', a)$ for all messages $m, m' \in \mathcal{M}$. The agent appending the message is called the *Adversary*, and is endowed

with a deterministic policy $f : \mathcal{S} \to \mathcal{M}$ to append messages $m = f(s)$ and an objective function $\mathcal{J}$ to optimise (details below).

The Victim is a standard reinforcement learning agent, selecting actions according to its policy $a_t \sim \pi_\theta(\cdot \mid s, f(s))$, where $a \in \mathcal{A}, s \in \mathcal{S}$. The Victim optimises its policy $\pi_\theta$ with respect to parameters $\theta$ in order to maximise its expected return $J$ as defined in Equation 1.

By contrast, the Adversary may only act by modifying the cheap talk channel features $f_\phi(s)$ at $s$ at every step, where $f_\phi : \mathcal{S} \to \mathcal{M}$ is a deterministic policy (function) of the current state and $\phi$ are the Adversary's parameters. These parameters may only be updated *between* full training/testing episodes of the Victim; the function remains static during episodes to avoid introducing non-stationarity and thus restrict the Adversary's range of influence. The Adversary's objective function $\mathcal{J}$ may be picked arbitrarily, and need not be differentiable if it is optimised using ES.

In our train-time experiments we focus on the fully-adversarial setting where objectives are zero-sum, $\mathcal{J} = -J$, and the allied setting where objectives are equal, $\mathcal{J} = J$. In test-time experiments we use an entirely different objective, such as reaching for an arbitrary circle in Reacher (see Figure 5c). This incentivises the Adversary to manipulate the Victim into maximising $\mathcal{J}$, even if this comes at the cost of the Victim's original objective $J$.

## 4.1 MINIMALITY OF CHEAP TALK MDPs

To justify our introductory claim that Cheap Talk MDPs only allow for a minimal range of influence, we first prove that Adversaries cannot influence Victims *whatsoever* in the tabular setting, irrespective of the Victim's learning algorithm. It follows that Adversaries can *only* attack Victims by interfering with their function approximator.

**Proposition 1.** *In any Cheap Talk MDP, the policy of a **tabular** Victim is independent from its Adversary provided uniform initialisation along $\mathcal{M}$, namely $\pi_0(\cdot \mid s_i, m_j) = \pi_0(\cdot \mid s_i, m_{j'}) \, \forall \, j, j'$.*

*Proof (Sketch).* The main intuition is that policy updates for different states do not interfere with each other in tabular settings. Assuming uniform initialisation and noticing that the only states encountered in the environment are of the form $(s, f(s))$, we deduce that the Victim's policy updates are independent from the Adversary's choice of function $f$. Formal proof in Appendix A.1. $\square$

We also prove more generally that Adversaries cannot prevent Victims with optimal convergence guarantees to converge to optimal rewards, even in non-tabular settings. They may however still harm the Victim by slowing down their convergence rate significantly.

**Proposition 2.** *A Victim which is **guaranteed to converge to optimal policies in MDPs** will also converge to optimal policies in Cheap Talk MDPs, with an expected return equal to the optimal return for the corresponding no-channel MDP.*

*Proof (Sketch).* Cheap Talk MDPs are just MDPs with augmented state spaces and transition / reward functions; a Victim will therefore converge regardless. Optimality of the expected return follows from the Bellman equation and independence of transition and reward functions from Adversaries. Formal proof in Appendix A.2. $\square$

Finally, we further justify minimality in Appendix A.3 by elaborating informally on the Adversaries' range of influence. We also discuss the possibility of further weakening Cheap Talk MDPs and conclude that all such variations either bring no advantage or reduce to regular MDPs.

## 5 METHOD

### 5.1 META-TRAINING PROCEDURE

Our method, Adversarial Cheap Talk (ACT), treats the problem setting as a meta-learning problem. The Adversary's parameters $\phi$ are only updated after *a full training (and testing) run* of the Victim's parameters $\theta$. In other words, $\phi$ is *static* during the whole training run (inner loop) of $\theta$ and only gets

updated once the inner loop completes, which prevents the introduction of non-stationarity. In the outer loop, we optimise the Adversary's objective $\mathcal{J}$ with respect to $\phi$ using ES as a black-box optimisation technique. Details, including the Cheap Talk channel sizes and the Victims hyperparameters can be found in Appendix E.

## 5.2 TRAIN-TIME INFLUENCE

When influencing train-time performance, we set $J$ to be the agent's mean reward throughout its entire training trajectory. We consider both "Adversarial" and "Allied" versions of ACT, whereby Adversaries try to minimise or maximise $J$ respectively ($\mathcal{J} = \pm J$). Pseudocode is provided in Algorithm 1, where $E$ is the number of Victim training episodes and $N$ is the ES population size.

## 5.3 TEST-TIME MANIPULATION

When manipulating test-time behaviour, the goal of the Adversary is to use the cheap talk features to maximise some arbitrary objective $\mathcal{J}$ during the Victim's test-time; however, the Adversary may also communicate messages during the Victim's training. Note that $\mathcal{J}$ can be *any* objective, including minimising or maximising the Victims-return. Because the train-time and test-time behaviour of the Adversary differ significantly,

---

**Algorithm 1** Train-time ACT

1: Set $c = \pm 1$ for allied / adversarial
2: Initialise Adversary parameters $\phi$
3: **for** $m = 0$ **to** $M$ **do**
4:     Sample $\phi_n \sim \phi + \sigma\epsilon_n$ with $\epsilon_n \sim \mathcal{N}(0, I)$
5:     **for** $n = 0$ **to** $N$ **do**
6:         Initialise Victim parameters $\theta$
7:         rewards = []
8:         **for** $e = 0$ **to** $E$ **do**
9:             s = env.reset()
10:             **while** not done **do**
11:                 $a \sim \pi_\theta(\cdot \mid s, f_{\phi_n}(s))$
12:                 $r, s$, done = env.step($a$)
13:                 rewards.append($r$)
14:             **end while**
15:             Update $\theta$ with PPO to maximise $J$
16:         **end for**
17:         $\mathcal{J}_n = c \cdot$ sum(rewards)/len(rewards)
18:     **end for**
19:     Update $\phi$ using ES to maximise $\mathcal{J}$
20: **end for**

---

we parameterise them separately (as $\phi$ and $\psi$ respectively), but optimise them jointly.

As an example, consider the Reacher environment (see 5c), where the Victim is trained to control a robot arm to reach for the blue circle. During the Victim's training, the train-time Adversary (parameterised by $\phi$) manipulates the cheap talk features to encode spurious correlations in the Victim's policy. At test-time, the test-time Adversary (parameterised by $\psi$) manipulates the cheap talk features to take advantage of the spurious correlations and control the Victim to have it reach for the yellow circle instead (the Adversary's objective $\mathcal{J}$). More concisely, the train-time Adversary wants to *create* a backdoor to make the Victim susceptible to manipulation at test-time. The test-time Adversary wants to *use* this backdoor to control the Victim. The train-time and test-time Adversaries ($\phi$ and $\psi$) are trained end-to-end to maximise $\mathcal{J}$. While such optimisation would be difficult for gradient-based methods due to the long-horizon nature of the problem, ES is agnostic to the length of optimisation horizon.

Note that the test-time Adversary $\psi$ only gets a single shot to maximise $\mathcal{J}$ at the end of the Victim's training and does not have access to (and thus cannot train against) the test-time parameters of the Victim $\theta'$. To describe this formally, let $\mathcal{T}(\theta \mid \phi)$ denote the distribution induced by the inner loop training with the train-time Adversary $\phi$ over Victim $\theta$. Then, in each meta-episode, the test-time Adversary $\psi$ has to interact with an unseen sample $\theta'$ of the distribution over trained Victims $\theta' \sim \mathcal{T}(\cdot \mid \phi)$. In Section 6, we show that the distribution $\mathcal{T}(\theta \mid \phi)$ has non-trivial variance, suggesting that it is difficult to train against. Moreover, in Figure 9 (Section 6), we provide empirical evidence that the train-time Adversary learns to reduce the variance of $\mathcal{T}(\theta \mid \phi)$ to help the test-time Adversary. Pseudocode is provided in Algorithm 2, Appendix B.

## 6 EXPERIMENTS AND RESULTS

We evaluate ACT on three different simple gym environments: Cartpole, Pendulum, and Reacher (Brockman et al., 2016). We also evaluate ACT on Minatar Breakout Young and Tian (2019); Lange (2022b) to test ACT's ability to scale to higher-dimensional environments. The Victim is trained with Proximal Policy Optimisation (Schulman et al., 2017, PPO). The Adversary is trained using ES (Salimans et al., 2017).

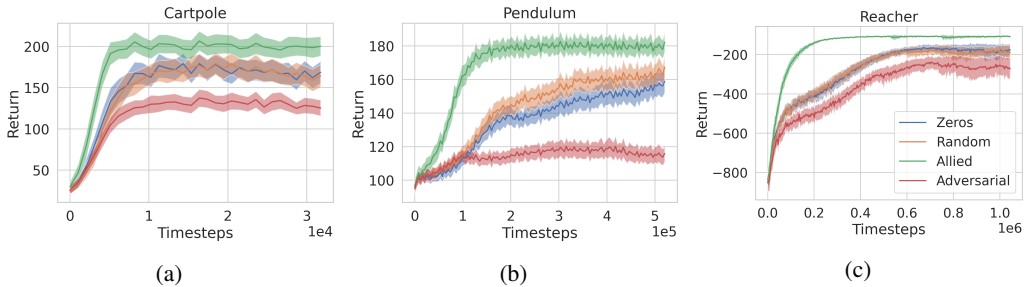

Figure 1: Visualisations of the training curves of the Victim across different Adversaries for (a) Cartpole, (b) Pendulum, and (c) Reacher. Error bars denote the standard error across 10 seeds of Victims trained against a single trained Adversary.

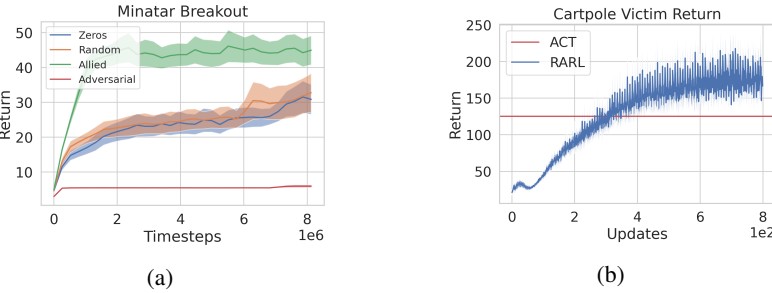

Figure 2: (a) Visualisations of the training curves of the Victim in Breakout-Minatar, a higher-dimensional environment, against different Adversaries. (b) Comparing Adversarial ACT to RARL. Eventually, the Victim learns to overcome the RARL adversary.

We train thousands of agents per minute on a single V100 GPU by vectorising both the *PPO algorithm itself* and the environments using Jax (Bradbury et al., 2018). This allows us to JIT-compile the *full training pipeline* and perform end-to-end deep RL training completely on GPUs. We adapt the environment implementations from Brockman et al. (2016) and Lenton et al. (2021) and use the ES implementation from Lange (2022a). This compute setup allows us to efficiently perform outer-loop ES on the full training trajectories of inner-loop PPO agents. For example, in Cartpole, we can simultaneously train 8192 PPO agents at a time on a single V100 GPU. Over 1024 generations of ES, this results in training 8,388,608 PPO agents from scratch in 2 hours on 4 V100 GPUs. The longest training time, which was the test-time Reacher setting, took 20 hours to train 1024 generations on 4 V100 GPUs. Training details are provided in Appendix E. We also include videos of the Victim's performance and visualisations of the Adversary's outputs in the Supplementary Materials.

## 6.1 TRAIN-TIME INFLUENCE

Figure 1 and Figure 2a show the results of training Victims alongside four different Adversaries.

1. **Ally**: meta-trained to *maximise* the Victim's mean reward throughout training.
2. **Adversary**: meta-trained to *minimise* the Victim's mean reward throughout training.
3. **Random Adversary**: randomly initialise and fix the Adversary's parameters $\phi$ using LeCun Uniform initialisation (LeCun et al., 2012).
4. **Zeroes Adversary**: appends only zeroes as messages.

**Ally.** The Ally manages to assist the Victim to learn and converge faster – this is likely done by appending useful features of the environment. We do further analysis in Appendix C Figure 7b.

**Adversary.** The Victims trained alongside the Adversaries are vastly outperformed by the baselines, even though the Adversary cannot change the dynamics of the underlying MDP, and cannot add

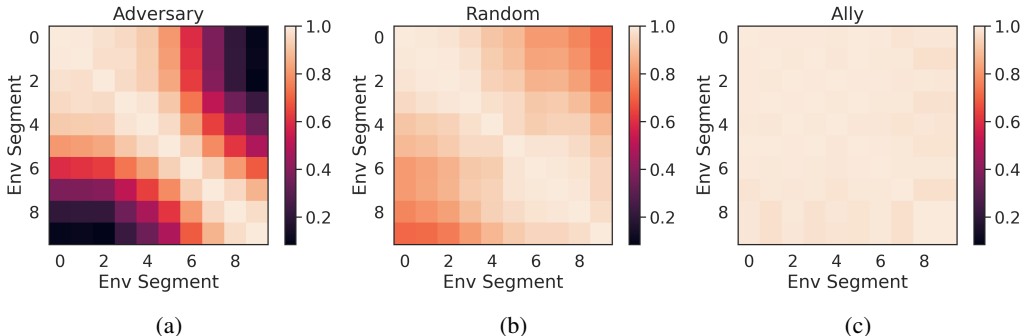

(a)  (b)  (c)

Figure 3: Visualisations of the cosine distance between gradient updates on different environment segments in cartpole. We collected each Victim's experience buffer before the agents converge in training and split each into 10 bins, ordered by the time-step within the environment. We then calculate the gradient update the agents perform on each of these bins. This is the technique used in Fedus et al. (2020). For the adversary (a), gradient updates on the early timesteps in an environment interfere with gradient updates on the ending timesteps. For the ally (c), they are positively correlated.

non-stationarity or stochasticity. Moreover, since Adversaries cannot influence tabular Victims by Proposition 1, this must be accomplished through learnt interference with the Victim's function approximator. We hypothesise that the Adversary may be inducing catastrophic interference within the environment, which was observed by Fedus et al. (2020) in Atari 2600 games. They show that features useful in the early phases of an environment episode can interfere with learning features for performing well in the later phases of an episode.

Figure 3 suggests that, in the Adversarial setting, the gradient updates performed for transitions sampled early in an episode can *interfere* with the gradient updates performed for transitions later in an episode. Meanwhile, those gradient updates are positively correlated in the Allied setting, suggesting that the gradient updates aid each other.

We also compare our evolutionary meta-optimisation procedure to Robust Adversarial Reinforcement Learning (RARL) (Pinto et al., 2017) in Figure 2b, which updates the adversary's parameters online using reinforcement learning. We include training details in Appendix F. In both settings, the Adversary can only communicate over the cheap talk channel; however, RARL is given a larger range of influence. Firstly, RARL introduces non-stationarity since it is updated online during the opponent's learning, unlike ACT. Secondly, RARL is parameterised by a stochastic policy, meaning that it can inject stochasticity into the environment, unlike ACT. Thirdly, RARL is able to train directly against the Victim's policy online, unlike ACT which cannot view the Victim's policy or actions. However, RARL ultimately underperforms ACT in the adversarial setting since it is unable to consider the long-term evolution of the opponent's policy (it takes greedy updates against the Victim's online parameters). Thus, the Victim learns to simply ignore the cheap talk channel. Note that RARL is *intended* to be overcome in order to robustify the Victim's policy.

## 6.2 Test-Time Manipulation

In test-time manipulation, the Adversary's objective is to maximise the score of the *goal-conditioned objectives* described in 4 at test-time. The Adversary needs to learn to introduce a backdoor during train-time and use the backdoor during test-time to fully control the Victim as explained in 5.3. To better understand the capability of our model, we investigate four different Adversary-Victim settings. These four settings serve as ablations to study the individual and joint performance of the train- and test-time Adversaries.

1. **Test-Time Adversary $\psi$ with Train-Time Adversary $\phi$:** This is the algorithm described in 2. First, we train a Victim $\theta$ alongside a train-time Adversary $\phi$. We then evaluate the return of the test-time Adversary $\psi$ according to the goal-conditioned return. Both the train- and test-time Adversaries are trained using ES. The test-time Adversary $\psi$ only gets a single shot against a Victim and is thus represented by a horizontal line in Figure 5.

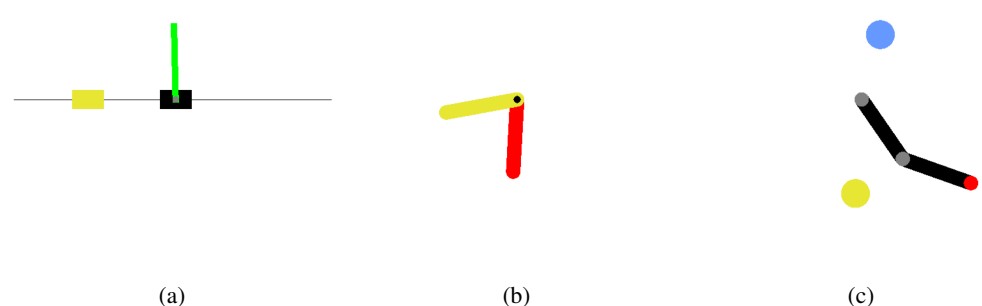

(a)                                    (b)                                    (c)

Figure 4: Visualisations of our goal-conditioned environments (a) In Cartpole, the Adversary's target is a randomly selected point on the x-axis (the yellow box). (b) In Pendulum, the Adversary's goal is a randomly selected angle (the yellow pole). (c) In Reacher, the Adversary's goal is a random point, (the yellow circle). The Victim's goal is the blue circle.

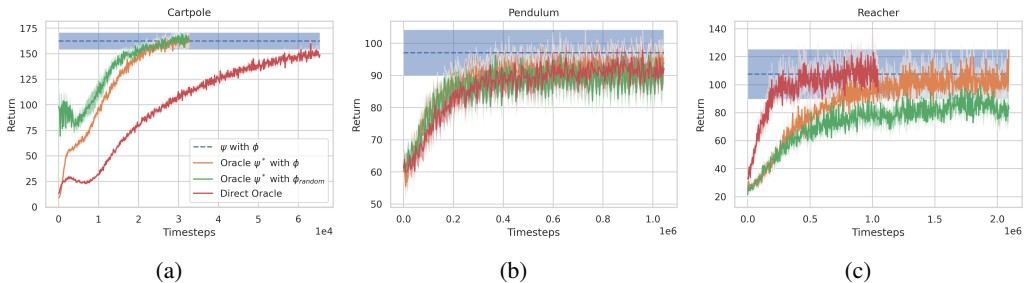

(a)                                    (b)                                    (c)

Figure 5: Training curves of the different agents in (a) Goal-Conditioned Cartpole (b) Goal-Conditioned Pendulum (c) Goal-Conditioned Reacher. The ablations show that the train- and test-time Adversary learn near-optimal performance in comparison to the oracles. Error bars denote the standard error of the mean across 10 seeds of Victim trained against a single trained Adversary.

2. **Test-Time Oracle $\psi^*$ with Train-Time Adversary $\phi$:** First, we optimise the Victim $\theta$ by training it alongside the above train-time Adversary $\phi$. Then, instead of ES, we use PPO to train the test-time Adversary $\psi^*$ against the Victim $\theta$. Unlike the test-time Adversary, the oracle $\psi^*$ is allowed to train against the pretrained and fixed Victim $\theta$ to maximise its returns, as described in Algorithm 4 in Appendix B.

3. **Test-Time Oracle $\psi^*$ with Random Train-Time Adversary $\phi_{\text{random}}$:** First, we obtain a Victim $\theta$ by training it alongside a *random* train-time Adversary, $\phi_{\text{random}}$, with randomly initialised and fixed parameters. Next, we use PPO to train the test-time Adversary $\psi^*$ to maximise the goal-conditioned return.

4. **Direct Oracle:** In this baseline, there is no cheap talk or Victim. We simply train a PPO agent to maximise the *goal-conditioned return $J$*. It can observe the full state and directly output actions in the environment.

All results are shown in Figure 5. We can compare **(1)** and **(4)** to measure how effective the train-time Adversary $\phi$ and test-time Adversary $\psi$ are at achieving the maximal possible return jointly. As Figure 5 shows, the train- and test-time Adversaries perform near-optimally.

By comparing **(2)** and **(3)**, we can observe how effective $\phi$ is at *shaping $\theta$*. In reacher, we can see that the test-time Oracle $\psi^*$ *cannot* achieve max performance with a random train-time Adversary.

We can compare **(1)** and **(2)** to see how effective the test-time Adversary $\psi$ is exploiting a given Victim $\theta$. We can see that $\psi$ achieves near-optimal performance even though it *has never trained against the specific Victim $\theta$ or had access to its parameters*. In Figure 9 we show that it can do this

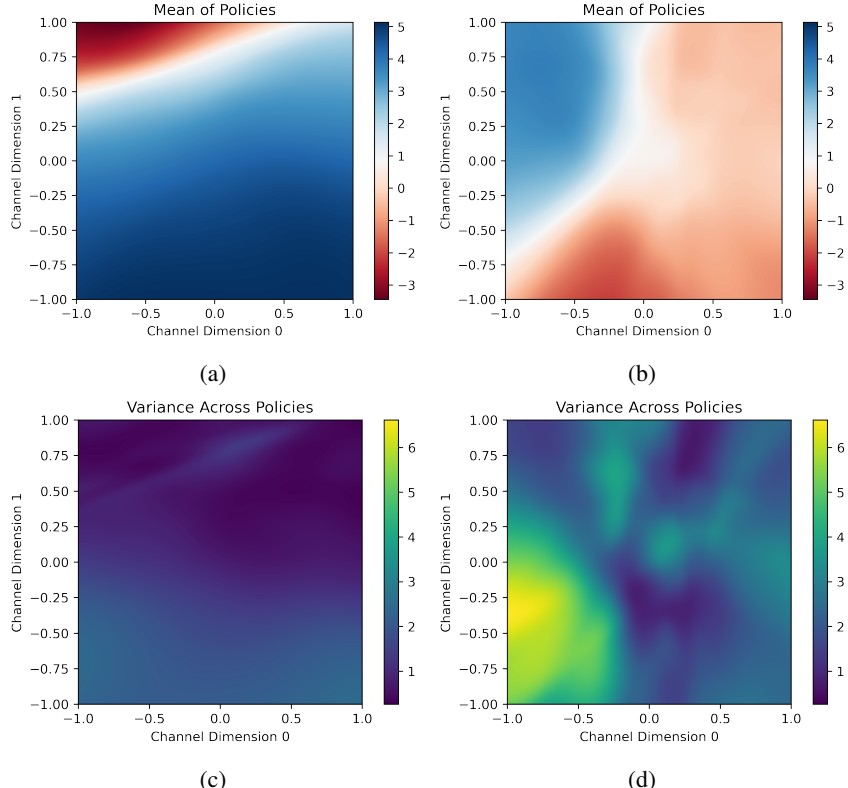

Figure 6: We train 10 different Victims alongside the Learned $\phi$ (a & c), and 10 different Victims alongside a randomly generated $\phi$ (b & d) in the Pendulum environment. (a) and (b) show the mean of the policy output across the 10 Victims as we vary the value of the message in a fixed randomly selected state. The policies trained with the learned $\phi$ achieve a much wider range of outputs. (c) and (d) show the variance of the policy output across the 10 Victims. The policies trained with the learned $\phi$ display very little variance.

because the train-time Adversary $\phi$ not only maximises the range of outputs that the cheap talk can achieve, but it also does so in a *consistent and low-variance* way.

## 7 CONCLUSION & FUTURE WORK

In this paper, we propose a novel, minimum-viable, adversarial setting for RL agents, where the Adversary can only influence the Victim over messages, and can only do so with a deterministic function that only depends on the current state. By training an Adversary with adversarial cheap talk (ACT), we show that appending to the observations of a learning agent, even with strong constraints, is sufficient to drastically improve or *decrease* a learning agent's train-time performance or introduce a backdoor to control the learning agent at test time completely. Furthermore, we provide an in-depth analysis of the behaviour of our Adversaries. At train-time, the Adversary learns to induce something akin to catastrophic interference to decrease training performance. In the instant test-time manipulation setting, the train-time Adversary learns to reduce the variance of the training process to reduce variance of potential Victims the test-time Adversary could face.

As RL models become more widespread, we believe practitioners should consider this new class of minimum viable attacks. Therefore, we propose using domain knowledge to identify and filter out potentially controllable information as the first defence measure. Identifying these channels without domain knowledge would be challenging: While there has been past work in identifying task-irrelevant features in reinforcement learning (Vischer et al., 2021), ACT features still contain task-relevant information since they are functions of the state. To defend against the test-time Adversary, one can detect when the input goes out-of-distribution (Lin et al., 2017). Future work would be required to build more robust and practical defences.

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

# A    Minimality of Cheap Talk MDPs

## A.1    Proof of Proposition 1

**Proposition 1.** *In any Cheap Talk MDP, the policy of a **tabular** Victim is independent from its Adversary provided uniform initialisation along $\mathcal{M}$, namely $\pi_0(\cdot \mid s_i, m_j) = \pi_0(\cdot \mid s_i, m_{j'}) \, \forall \, j, j'$.*

*Proof.* In a Cheap Talk MDP $\langle \mathcal{S}, \mathcal{A}, \mathcal{P}, \mathcal{R}, \gamma, \mathcal{M}, f, \mathcal{J} \rangle$, a tabular Victim arbitrarily orders states as $\{s_1, \ldots, s_d\}$ and messages as $\{m_1, \ldots, m_k\}$, where $d = |\mathcal{S}|$ and $k = |\mathcal{M}|$, and stores policies $\pi_t(\cdot \mid s_i, m_j)$ at time $t$ of the learning process for all $i \in [d], j \in [k]$. The argument follows identically for value functions. Assuming uniform initialisation along the $\mathcal{M}$ axis means that

$$\pi_0(\cdot \mid s_i, m_j) = \pi_0(\cdot \mid s_i, m_{j'})$$

for all $j, j' \in [k]$. Now consider any two Adversaries $f, g$ and their influence on two copies of the same Victim $V, W$ with respective policies $\pi, \chi$. The only states encountered in the environment are of the form $(s, f(s))$ and $(s, g(s))$ respectively, so Victims only update the corresponding policies

$$\pi_t(\cdot \mid s_i, f(s_i)) \qquad \text{and} \qquad \chi_t(\cdot \mid s_i, g(s_i)).$$

We prove by induction that these quantities are equal for all $t$. The base case holds by uniform initialisation along $\mathcal{M}$; assume the claim holds for all fixed $0 \le t \le T$. The Victims update their policies at time $T+1$ according to the same learning rule, as a function of the transitions and returns under current and past policies $\pi_t$ and $\chi_t$ respectively. Transitions take the form $(s, f(s), a, s', f(s'))$ for $V$ and $(s, g(s), a, s', g(s))$ for $W$, which have identical probabilities and returns because

$$\pi_t(a \mid s_i, f(s_i)) = \chi_t(a \mid s_i, g(s_i));$$
$$\mathcal{P}(s', f(s') \mid s, f(s), a) = \mathcal{P}(s', g(s') \mid s, g(s), a);$$
$$\mathcal{R}(s, f(s), a) = \mathcal{R}(s, g(s), a)$$

by inductive assumption and independence of $\mathcal{P}, \mathcal{R}$ from $\mathcal{M}$. This implies that the Victims' policies $\pi_T(\cdot \mid s_i, f(s_i)) = \chi_T(\cdot \mid s_i, g(s_i))$ are updated identically to

$$\pi_{T+1}(\cdot \mid s_i, f(s_i)) = \chi_{T+1}(\cdot \mid s_i, g(s_i))$$

as required to complete induction. Note that this would not necessarily hold in non-tabular settings, where updating parameters $\theta$ of the function approximator for some state $(s_i, f(s_i))$ may alter the policy on some other state $(s_j, f(s_j))$. It now follows that trajectories $\tau = (s^k, f(s^k), a^k)_k$ for $V$ and $\omega = (s^k, g(s^k), a^k)_k$ for $W$ have identical probabilities and hence produce identical returns

$$\mathbb{E}_{\tau \sim \pi_t} [\mathcal{R}(\tau)] = \mathbb{E}_{\omega \sim \chi_t} [\mathcal{R}(\omega)]$$

at any timestep $t$ of the learning process, concluding independence from Adversaries. $\qquad \square$

## A.2    Proof of Proposition 2

**Proposition 2.** *A Victim which is **guaranteed to converge to optimal policies in MDPs** will also converge to optimal policies in Cheap Talk MDPs, with an expected return equal to the optimal return for the corresponding no-channel MDP.*

*Proof.* By assumption, the Victim is guaranteed to converge to an optimal policy $\bar{\pi}$ in any given Cheap Talk MDP $\langle \mathcal{S}, \mathcal{A}, \mathcal{P}, \mathcal{R}, \mathcal{M}, f, \mathcal{J}, \gamma \rangle$, since a Cheap Talk MDP is itself an MDP with an augmented state space $\mathcal{S} \times \mathcal{M}$ and augmented transition/reward functions that are defined to be independent from $\mathcal{M}$. Now $\bar{\pi}$ naturally induces a policy $\pi$ on the no-channel MDP, given by $\pi(\cdot \mid s) := \bar{\pi}(\cdot \mid s, f(s))$, and in particular $Q(s, a) = \bar{Q}(s, f(s), a)$ by independence of transitions and rewards from $\mathcal{M}$. Optimality of $\pi$ follows directly from the Bellman equation

$$Q(s, a) = \bar{Q}(s, f(s), a) = \mathbb{E}_{s' \sim \mathcal{P}(\cdot \mid s, a), r \sim \mathcal{R}(\cdot \mid s, a)} \left[ r + \gamma \max_{a' \in \mathcal{A}} \bar{Q}(s', f(s'), a') \right]$$

$$= \mathbb{E}_{s' \sim \mathcal{P}(\cdot \mid s, a), r \sim \mathcal{R}(\cdot \mid s, a)} \left[ r + \gamma \max_{a' \in \mathcal{A}} Q(s', a') \right].$$

Now trajectories $\bar{\tau} = (s^k, f(s^k), a^k)_k$ and $\tau = (s^k, a^k)_k$ have identical probability and return under $\pi$ and $\bar{\pi}$ respectively, so the Victim has expected return

$$\mathbb{E}_{\bar{\tau} \sim \bar{\pi}} [\mathcal{R}(\bar{\tau})] = \mathbb{E}_{\tau \sim \pi} [\mathcal{R}(\tau)]$$

which is the optimal expected return of the original no-channel MDP. $\qquad \square$

## A.3 FURTHER INFORMAL DISCUSSION

Consider a Cheap Talk MDP $\langle \mathcal{S}, \mathcal{A}, \mathcal{P}, \mathcal{R}, \gamma, \mathcal{M}, f, \mathcal{J} \rangle$. For a fixed training / testing run of the Victim on the MDP, the Adversary outputs a message $f(s)$ at each step according to a fixed deterministic function $f : \mathcal{S} \rightarrow \mathcal{M}$. Below we elaborate informally on the claims that Adversaries cannot (1) occlude the ground truth, (2) influence the environment dynamics / reward functions, (3) see the Victim's actions or parameters, (4) inject stochasticity, or (5) introduce non-stationarity.

(1) The message is *appended* to the state $s$ and the Victim acts with full visibility of the ground truth (state) $s$ according to its policy: $a \sim \pi(\cdot \mid s, f(s))$.

(2) The transition and reward functions $\mathcal{P}, \mathcal{R}$ are defined to be independent from $\mathcal{M}$. Formally we have $\mathcal{P}(\cdot \mid s, m, a) = \mathcal{P}(\cdot \mid s, m', a)$ for all $m, m' \in \mathcal{M}$ (similarly for $\mathcal{R}$), so the Adversary's choice of message $m = f(s)$ cannot influence $\mathcal{P}$ or $\mathcal{R}$.

(3) $f : \mathcal{S} \rightarrow \mathcal{M}$ is defined as a function of $\mathcal{S}$ only, so the Adversary cannot condition its policy based on the Victim's actions or parameters (i.e. it cannot see or influence them).

(4) $f$ is a deterministic function, so $\pi(\cdot \mid s, f(s))$ is a distribution only on actions $\mathcal{A}$. The transition and reward functions are independent from $f$, so they are distributions only on state-action pairs $\mathcal{S} \times \mathcal{A}$. It follows that the Adversary injects no further stochasticity into the MDP.

(5) $f$ is static for a fixed training / testing run, so $s_t = s_{t'}$ implies $f(s_t) = f(s_{t'})$ for all timesteps $t, t'$ in the run. It follows that any given Victim policy $\pi$ is stationary, namely $\pi(\cdot \mid s_t, f(s_t)) = \pi(\cdot \mid s_{t'}, f(s_{t'}))$ for all $s_t = s_{t'}$. Since $\mathcal{P}$ and $\mathcal{R}$ are stationary (as defined by a standard MDP) and independent from $\mathcal{M}$, their stationarity is also preserved.

Finally, we discuss the possibility of further weakening components of a Cheap Talk MDP, and conclude that all such variants (A-E) bring no advantage or reduce to regular MDPs.

(A) Removing the channel $\mathcal{M}$ or the policy $f : \mathcal{S} \rightarrow \mathcal{M}$ would result in the Victim being completely independent from the Adversary, so no adversarial influence could be exerted whatsoever.

(B) Restricting the capacity of $\mathcal{M}$ to a certain number of bits would further restrict an Adversary's range of influence, so one could say that the *truly* minimum-viable setting is to impose a set of size $|\mathcal{M}| = 1$. However, cheap talk is still cheap talk when varying capacity, and there is no reason to arbitrarily restrict the size to 1 if we are to apply our setting to complex environments likely requiring more than a single bit of communication to witness interesting results.

(C) Not allowing Adversaries to see states, namely removing $\mathcal{S}$ as inputs to $f$, yields a function $f : \{0\} \rightarrow \mathcal{M}$ which always outputs the same message $f(0) = m \in \mathcal{M}$. This is equivalent to the previous restriction of imposing a set $\mathcal{M}$ of size 1, since in this case any function $f : \mathcal{S} \rightarrow \mathcal{M}$ would have to output the unique element $f(s) = m$ for all input states $s$.

(D) The Adversary must have some objective function $\mathcal{J}$ in order for an adversarial setting to make sense – removing it would remove the Adversary's reason to exist, since it would have no incentive to learn parameters that influence the Victim according to some goal.

(E) Restricting the function class of objectives $\mathcal{J}$ is a valid minimisation of the setting, but simply restricts our interest in the setting itself. The setting should at the very least allow for adversarial objectives of the form $\mathcal{J} = \pm J$, as we consider in the train-time setting. In test-time, our aim is to show how Adversaries can exert arbitrary control over Victims despite cheap talk restrictions, and we therefore consider more general objective functions.

## B  PSEUDOCODE

---

**Algorithm 2** Test-time ACT

---

1: Initialize train-time ACT parameters $\phi$
2: Initialize test-time ACT parameters $\psi$
3: **for** $m = 0$ **to** $M$ **do**
4:    Sample $\phi_n \sim \phi + \sigma\epsilon_n$ where $\epsilon_1, ..., \epsilon_N \sim \mathcal{N}(0, I)$
5:    Sample $\psi_n \sim \psi + \sigma\epsilon_n$ where $\epsilon_1, ..., \epsilon_N \sim \mathcal{N}(0, I)$
6:    **for** $n = 0$ **to** $N$ **do**
7:       Initialize policy params $\theta$
8:       rewards = []
9:       **for** $e = 0$ **to** $E$ **do**
10:          s = env.reset()
11:          **while** not done **do**
12:             $m = f_{\phi_n}(s)$
13:             $\bar{s} = [s, m]$
14:             $a \sim \pi_\theta(\cdot \mid \bar{s})$
15:             $r, s$ = env.step($a$)
16:          **end while**
17:          Update $\theta$ using PPO to maximise $J$
18:       **end for**
19:       **for** $i = 0$ **to** $I$ **do**
20:          s = env.reset()
21:          **while** not done **do**
22:             $m = f_{\psi_n}(s)$
23:             $\bar{s} = [s, m]$
24:             $a \sim \pi_\theta(\cdot \mid \bar{s})$
25:             $r, s$, done = env.step($a$)
26:             $r_t^S = R^S(s, a)$
27:             rewards.append($r_t^S$)
28:          **end while**
29:       **end for**
30:    **end for**
31:    Update $\phi$ using ES to maximise $\mathcal{J}$
32:    Update $\psi$ using ES to maximise $\mathcal{J}$
33: **end for**

---

---

**Algorithm 3** Test-time Oracle PPO ACT

---

1: Initialize train-time ACT parameters $\phi$
2: Obtain trained $\phi, \theta$ from Algorithm 2
3: Initialize test-time ACT parameters $\psi^*$
4: **for** $i = 0$ **to** $I$ **do**
5:    s = env.reset()
6:    **while** not done **do**
7:       $m \sim \pi_{\psi^*}(\cdot \mid s)$
8:       $\bar{s} = [s, m]$
9:       $a \sim \pi_\theta(\cdot \mid \bar{s})$
10:      $r, s, \text{done} = \text{env.step}(a)$
11:      $r_t^S = R^S(s, a)$
12:      rewards.append($r_t^S$)
13:    **end while**
14:    Update $\psi^*$ using PPO to maximise $\mathcal{J}$
15: **end for**

---

---

**Algorithm 4** Test-time Random Shaper

---

1: Initialize train-time ACT parameters $\phi_{\text{random}}$
2: Initialize policy params $\theta$
3: rewards = []
4: **for** $e = 0$ **to** $E$ **do**
5:     s = env.reset()
6:     **while** not done **do**
7:        $m = f_{\phi_{\text{random}}}(s)$
8:        $\bar{s} = [s, m]$
9:        $a \sim \pi_\theta(\cdot \mid \bar{s})$
10:       $r, s = \text{env.step}(a)$
11:     **end while**
12:     Update $\theta$ using PPO to maximise $J$
13: **end for**
14: Initialize test-time ACT parameters $\psi^*$
15: **for** $i = 0$ **to** $I$ **do**
16:     s = env.reset()
17:     **while** not done **do**
18:        $m \sim \pi_{\psi^*}(\cdot \mid s)$
19:        $\bar{s} = [s, m]$
20:        $a \sim \pi_\theta(\cdot \mid \bar{s})$
21:        $r, s = \text{env.step}(a)$
22:        $r_t^S = R^S(s, a)$
23:        rewards.append($r_t^S$)
24:     **end while**
25:     Update $\psi^*$ using PPO to maximise $\mathcal{J}$
26: **end for**

---

# C  ABLATIONS

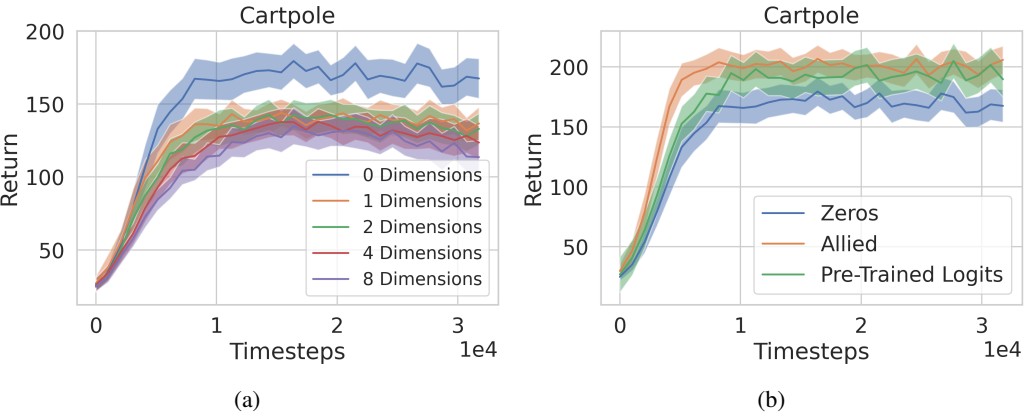

(a)               (b)

Figure 7: (a) Ablations on the different number of cheap talk dimensions for the Adversary in Cartpole. We find that for a low-dimensional environment like Cartpole, the Adversary does not achieve much marginal improvement from increasing the number of channels, suggesting that there may be some limit to the amount that it can harm performance. (b) Comparing the ally with an Adversary that outputs pre-trained logits in Cartpole. We find that the allied ACT still performs better, implying that it is outputting features that are more useful than logits from a pre-trained policy. Error bars denote the standard error across 10 seeds of a Victim trained against a single meta-trained Adversary.

# D    PENDULUM ABLATION

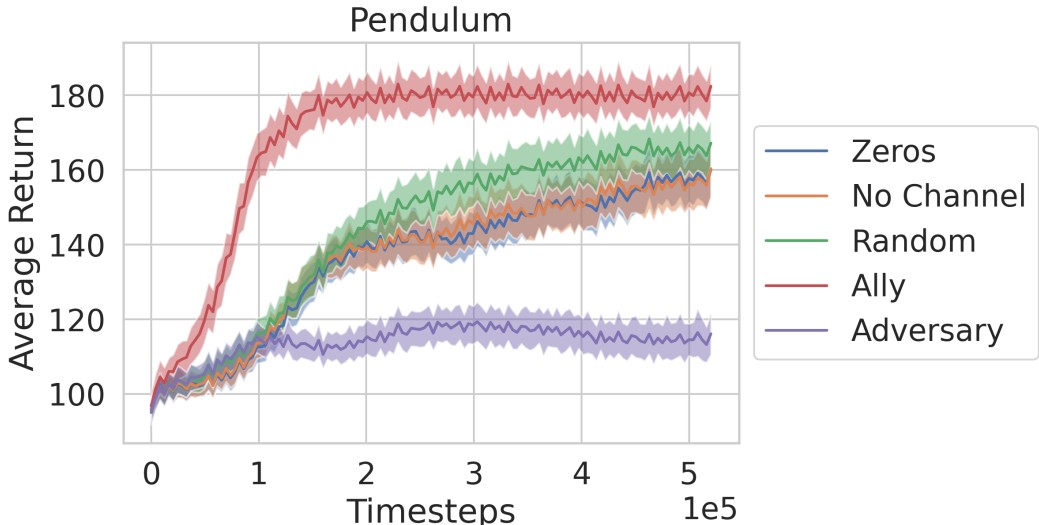

Figure 8: Interestingly, it seems like random network features improved performance in Pendulum. To make sure this was not due to network initialisation effects, we ran an ablation where we removed the cheap talk channel. It achieves about the same performance as a channel with zeros, which implies that the performance difference is not due to network initialisation.

## E  HYPERPARAMETER DETAILS

We report the hyperparameter values used for each environment in our experiments. Our PPO implementation uses observation normalisation (a common design choice for PPO), which means that the attack is invariant to the *range* of values outputted over the cheap talk channel.

Table 1: Important parameters for the Cartpole environment

| Parameter | Value |
|---|---|
| State Size | 4 |
| message Size | 2 |
| message Range | $-2\pi, 2\pi$ |
| Number of Environments | 4 |
| Maximum Grad Norm | 0.5 |
| Number of Updates | 32 |
| Update Period | 256 |
| Outer Discount Factor $\gamma$ | 0.99 |
| Number of Epochs per Update | 16 |
| PPO Clipping $\epsilon$ | 0.2 |
| General Advantage Estimation $\lambda$ | 0.95 |
| Critic Coefficient | 0.5 |
| Entropy Coefficient | 0.01 |
| Learning Rate | 0.005 |
| Population Size | 1024 |
| Number of Generations | 2049 |
| Outer Agent (OA) Hidden Layers | 2 |
| OA Size of Hidden Layers | 64 |
| OA Hidden Activation Function | ReLU |
| OA Output Activation Function | Tanh |
| Inner Agent (IA) Actor Hidden Layers | 2 |
| IA Size of Actor Hidden Layers | 32 |
| IA Number of Critic Hidden Layers | 2 |
| IA Size of Critic Hidden Layers | 32 |
| IA Activation Function | Tanh |
| Number of Rollouts | 4 |

Table 2: Important parameters for the Pendulum environment

| Parameter | Value |
| --- | --- |
| State Size | 3 |
| message Size | 2 |
| message Range | $-2\pi, 2\pi$ |
| Number of Environments | 16 |
| Maximum Grad Norm | 0.5 |
| Number of Updates | 128 |
| Update Period | 256 |
| Outer Discount Factor $\gamma$ | 0.95 |
| Number of Epochs per Update | 16 |
| PPO Clipping $\epsilon$ | 0.2 |
| General Advantage Estimation $\lambda$ | 0.95 |
| Critic Coefficient | 0.5 |
| Entropy Coefficient | 0.005 |
| Learning Rate | 0.02 |
| Population Size | 768 |
| Number of Generations | 2049 |
| Outer Agent (OA) Hidden Layers | 2 |
| OA Size of Hidden Layers | 64 |
| OA Hidden Activation Function | ReLU |
| OA Output Activation Function | Tanh |
| Inner Agent (IA) Actor Hidden Layers | 1 |
| IA Size of Actor Hidden Layers | 32 |
| IA Number of Critic Hidden Layers | 1 |
| IA Size of Critic Hidden Layers | 32 |
| IA Activation Function | Tanh |
| Number of Rollouts | 4 |

Table 3: Important parameters for the Reacher environment

| Parameter | Value |
| --- | --- |
| State Size | 10 |
| message Size | 4 |
| message Range | $-2\pi, 2\pi$ |
| Number of Environments | 32 |
| Maximum Grad Norm | 0.5 |
| Number of Updates | 256 |
| Update Period | 128 |
| Outer Discount Factor $\gamma$ | 0.99 |
| Number of Epochs per Update | 10 |
| PPO Clipping $\epsilon$ | 0.2 |
| General Advantage Estimation $\lambda$ | 0.95 |
| Critic Coefficient | 0.5 |
| Entropy Coefficient | 0.0005 |
| Learning Rate | 0.004 |
| Population Size | 128 |
| Number of Generations | 2049 |
| Outer Agent (OA) Hidden Layers | 2 |
| OA Size of Hidden Layers | 64 |
| OA Hidden Activation Function | ReLU |
| OA Output Activation Function | Tanh |
| Inner Agent (IA) Actor Hidden Layers | 2 |
| IA Size of Actor Hidden Layers | 128 |
| IA Number of Critic Hidden Layers | 2 |
| IA Size of Critic Hidden Layers | 128 |
| IA Activation Function | ReLU |
| Number of Rollouts | 4 |

Table 4: Important parameters for the Minatar environments

| Parameter | Value |
|---|---|
| State Size | 400 |
| message Size | 32 |
| message Range | $-2\pi, 2\pi$ |
| Number of Environments | 64 |
| Maximum Grad Norm | 0.5 |
| Number of Updates | 1024 |
| Update Period | 256 |
| Outer Discount Factor $\gamma$ | 0.99 |
| Number of Epochs per Update | 32 |
| PPO Clipping $\epsilon$ | 0.2 |
| General Advantage Estimation $\lambda$ | 0.95 |
| Critic Coefficient | 0.5 |
| Entropy Coefficient | 0.01 |
| Learning Rate | 3e-4 |
| Population Size | 128 |
| Number of Generations | 256 |
| Outer Agent (OA) Hidden Layers | 2 |
| OA Size of Hidden Layers | 64 |
| OA Hidden Activation Function | ReLU |
| OA Output Activation Function | Tanh |
| Inner Agent (IA) Actor Hidden Layers | 2 |
| IA Size of Actor Hidden Layers | 256 |
| IA Number of Critic Hidden Layers | 2 |
| IA Size of Critic Hidden Layers | 256 |
| IA Activation Function | ReLU |
| Number of Rollouts | 1 |

# F RARL HYPERPARAMETER DETAILS

Table 5: RARL Cartpole Parameters

| Parameter | Value |
|---|---|
| State Size | 4 |
| message Size | 2 |
| message Range | $-2\pi, 2\pi$ |
| Maximum Grad Norm | 0.5 |
| Total Number of Adversary and Learner Updates | 100 |
| Number of Learner Update Steps per Adversary Update | 8 |
| Number of Adversary Update Steps per Learner Update | 8 |
| Update Period | 256 |
| Outer Discount Factor $\gamma$ | 0.99 |
| Number of Epochs per Update | 16 |
| PPO Clipping $\epsilon$ | 0.2 |
| General Advantage Estimation $\lambda$ | 0.95 |
| Critic Coefficient | 0.5 |
| Entropy Coefficient | 0.01 |
| Learning Rate | 0.005 |
| Population Size | 1024 |
| Number of Generations | 2049 |
| Outer Agent (OA) Hidden Layers | 2 |
| OA Size of Hidden Layers | 64 |
| OA Hidden Activation Function | ReLU |
| OA Output Activation Function | Tanh |
| Inner Agent (IA) Actor Hidden Layers | 2 |
| IA Size of Actor Hidden Layers | 32 |
| IA Number of Critic Hidden Layers | 2 |
| IA Size of Critic Hidden Layers | 32 |
| IA Activation Function | Tanh |
| Number of Rollouts | 16 |

# G EXTRA VISUALISATIONS

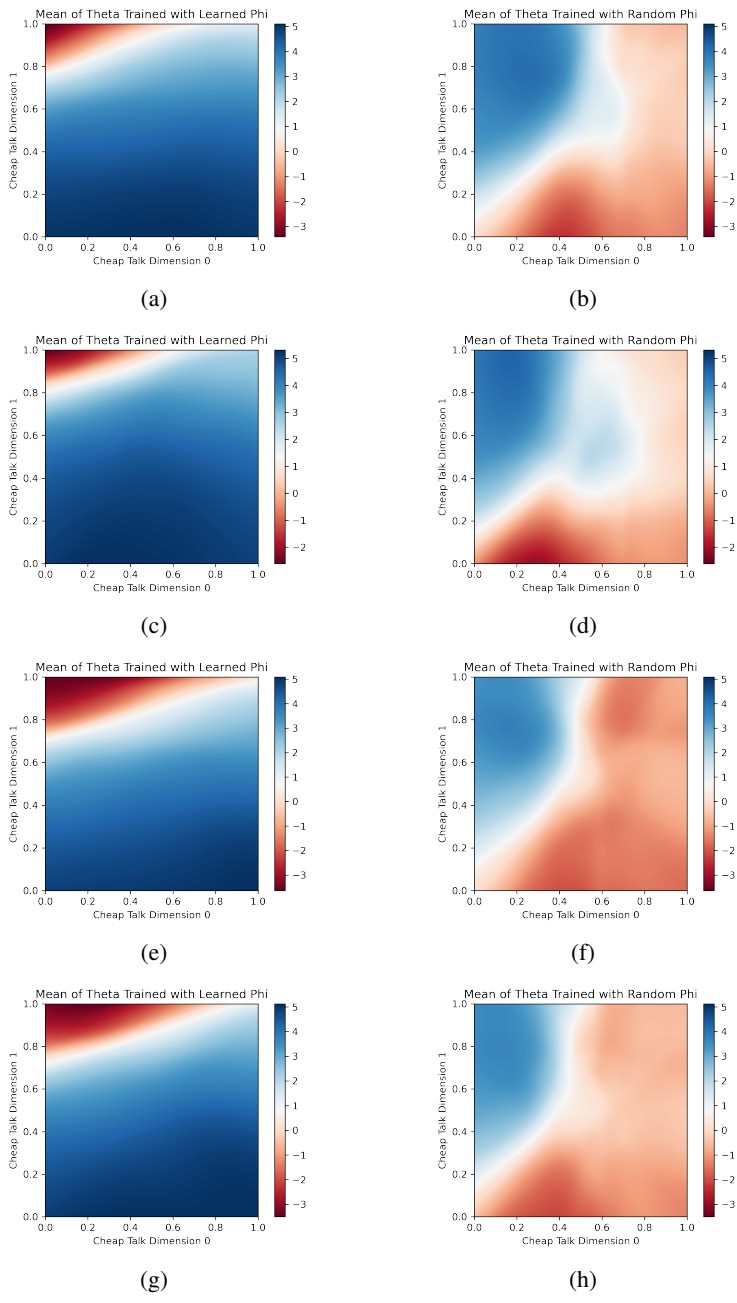

Figure 9: We train 10 different Victims alongside the Learned $\phi$ (left column), and 10 different Victims alongside a randomly generated $\phi$ (right column) in the Pendulum environment. We show the mean of the policy output across the 10 Victims as we vary the value of the message in multiple randomly selected states. The learned $\phi$ consistently generates similar policy outputs across different states with respect to the cheap talk channel.

## H   ADDITIVE PERTURBATIONS

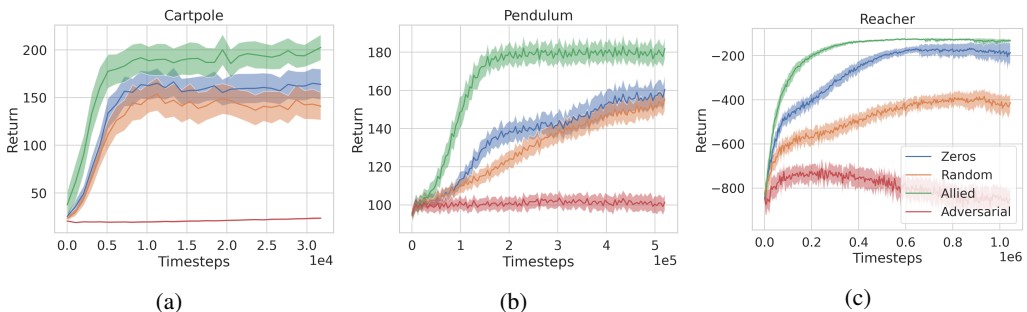

(a)                                    (b)                                    (c)

Figure 10: Visualisations of the training curves of the Victim across different Adversaries for (a) Cartpole, (b) Pendulum, and (c) Reacher. Error bars denote the standard error across 10 seeds of Victims trained against a single trained Adversary. In this setting, the Adversary *adds* the perturbation to the input rather than appending. Note that this allows the Adversary to conflate states and influence the optimal policy. Thus, the Adversary can harm performance far more.

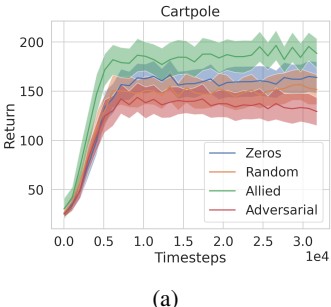

(a)

Figure 11: Visualisations of the training curves of the Victim across different Adversaries for Cartpole. Error bars denote the standard error across 10 seeds of Victims trained against a single trained Adversary. In this setting, the Adversary *adds* the perturbation to the useless features identified in (Vischer et al., 2021) rather than appending. It achieves similar performance to the cheap talk channel attacks.

