# OpenReview forum: "Adversarial Cheap Talk"
_ICLR.cc/2023/Conference — Submitted to ICLR 2023_

### Official Review · Reviewer_fsi4 · 2022-10-24

**Confidence:** 4
**Clarity, Quality, Novelty And Reproducibility:** The paper is very clearly written.
**Correctness:** 3
**Technical Novelty And Significance:** 4
**Empirical Novelty And Significance:** 4
**Recommendation:** 8

**Strength And Weaknesses:**

Strengths

While there has been extensive work on adversarial attacks in pattern recognition tasks, less work has been devoted overall to the problem of adversarial attacks in reinforcement learning. RL is particularly interesting for adversarial attacks because there are multiple potential attack points, beyond the input to the system itself, such as the policy, value computation, etc.

Most approaches in RL attacks require significant knowledge about the algorithm or the victim. This study focuses on the more challenging (and arguably realistically more critical) form of black box attacks.

Fig. 5 is impressive in showing results that are near the oracle predictions for the proposed adversaries.

Weaknesses

Rather than a weakness, this is a strength and a question. If I interpret the results correctly, the results for the proposed algorithm in the minatar breakout are particularly devastating. I was somehow expecting the results to be weaker than those in Fig. 1 given the increased complexity of the task. I guess that it is hard to compare quantitatively across completely different tasks and reward values, but is there any intuition for why the algorithm should work perhaps better in Fig. 2a? Is it somehow easier to attack minatar breakout?

Most of the remaining figures focus on the simpler tasks. It would be interesting if possible to expand some of the other figures (goal conditioning, comparison to oracle, showing policies) for the more complex breakout task.



**Summary Of The Paper:**

This study proposes an innovative strategy to attack reinforcement learning algorithms. Importantly, the attack requires minimal, information about the victim, in contrast to many other approaches that assume much more extensive inside information. Such black box approaches are necessarily less effective but ultimately much more dangerous because they can be applied in a generic fashion.

**Summary Of The Review:**

This paper addresses a timely and interesting problem in ML, namely, how to perform adversarial attacks for RL with minimal, if any, information about the victims. The paper is clearly written and shows very promising results in this direction, surpassing benchmarks and introducing a new methodology for black box RL attacks.

---

> ### Author Response · Authors · 2022-11-14
> **Response to Reviewer fsi4**
>
> We would like to thank Reviewer fsi4 for their clear and helpful feedback! We are glad that the reviewer finds the paper “timely”, “interesting”, and “promising”! The reviewer brought up good questions and suggestions that we would like to discuss below.
>
> >is there any intuition for why the algorithm should work perhaps better in Fig. 2a? Is it somehow easier to attack minatar breakout?
>
> This is a good question! Indeed, the results seem counterintuitive at first glance. We believe there are a couple of reasons why it is easier to attack Minatar Breakout:
>
> 1. Because the observation space is much larger, it is far more challenging for the PPO agent to learn appropriate features. In particular, each dimension in the observation does not necessarily correlate strongly with the reward or optimal policy. This makes it much easier for the Adversary to generate stronger adversarial correlations (relative to the rest of the observation dimensions), which can cause the function approximator to converge on the cheap talk channel inputs quickly.
>
> 2. Minatar Breakout also has relatively sparser rewards than the other settings we studied (which are fairly dense). This makes it a more difficult exploration problem, and the cheap talk channel can encode correlations that influence how the Victim explores.
>
> >It would be interesting if possible to expand some of the other figures (goal conditioning, comparison to oracle, showing policies) for the more complex breakout task.
>
> This is a fair point! We are currently investigating experiments where we have the test-time adversary optimise a return-conditioned goal. However, because of our limited compute budget and the fact that Minatar Breakout takes longer to train, we are prioritising other experiments. We hope to have these results by the end of the rebuttal period, but we do not expect them to change the main takeaways from the paper significantly.

---

### Official Review · Reviewer_HHmW · 2022-11-01

**Confidence:** 5
**Correctness:** 3
**Technical Novelty And Significance:** 4
**Empirical Novelty And Significance:** 4
**Recommendation:** 5

**Clarity, Quality, Novelty And Reproducibility:**

Clarity and Quality: In general, the paper is easy to understand at a high level. However, several details are not presented which affects my final rating:
1. Does the attack work for only on-policy learning algorithms or it generalizes to also off-policy algorithms? It would be good to make the scope of the attack clear.

2. What are the useless features of the learning environments (e.g. Cartpole, Pendulum)? Are they part of the state space of the standard agent without attack or new features are added by the authors? If the former, how did the authors determine the attack subset for different environments? If the latter, then why does the addition of these extra features make sense?

3. Since the messages add numerical values to the state, are these numerical values outliers or similar to values for the useful features? Why did you not constrain the length of the message as well as the range of values appearing in the appended messages?

4. For figure 3, the author mention "shows the cosine distance between these gradient updates." how are the gradient updates shown on the x-axis and y-axis different?

5. In Figure 2 (b), the number of updates shown on the x-axis seems small to make any conclusion about the trends for RARL and ACT?

6. Does proposition 1 hold for any tabular RL algorithm?

7. "is thus represented by a horizontal line in Figure 5." Algorithm 2, generates different $\psi$ for different iterations. Which $\psi$ is used?

8. Why are the number of timesteps in Fig 1 and 5 for different environments different?

9. What is the runtime of training and test time attacks?

10. Also, the algorithms require training large number of agents for different ES iterations. How realistic is this assumption?

Novelty: The threat model presented here is novel. The algorithms build upon ES which is a known technique, however its use for training and testing time attack on RL is new. The authors should consider providing more details about how ES learns $\phi$ and $\psi$ instead of treating it like black-box.

Reproducibility: The details about which and how messages are added may prevent reproducibility. The authors do not mention if they will make code available.

**Strength And Weaknesses:**

Strengths:
1. There do not exist enough studies on the vulnerability of deep reinforcement learning algorithms against training time attacks. New attacks as designed here can promote the development of more robust algorithms.

2. The attack model presented here makes several black-box assumptions that make the attack more feasible.

Weaknesses:
1. The threat model does not limit the size of the message appended by the adversary.

2. The evaluation is performed only on PPO. However, better algorithms like TD3, and SAC also exist. It is not clear if the proposed attack can affect learning by these algorithms.

3. Experimental details are not clear (as described below).

**Summary Of The Paper:**

This paper considers train and test time adversarial attacks on deep reinforcement learning. Here, the attacker can append the state observed by the learning agent with messages given by a deterministic policy. In the threat model, the attacker does not influence transition dynamics or reward functions, introduce non-stationarity, or access agents' actions or parameters. The authors develop both types of attacks based on evaluation strategies (ES). The experimental evaluation considers training with PPO for popular gym environments as well as minatar.  The results show that the training-time attacks can interfere with the learning of the neural network used by the agent while test-time attacks provide direct control over the agent.

**Summary Of The Review:**

Overall, the paper tackles a less explored but important problem that can promote the development of more robust RL algorithms. However, I found that the threat model and experimental details are not clearly presented due to which I consider the paper below the acceptance threshold for ICLR.

---

> ### Author Response · Authors · 2022-11-14
> **Response to Reviewer HHmW**
>
> We would like to thank Reviewer HHmW for their very detailed review! We are glad that the reviewer finds the work  “important” and “new”. Using the reviewer’s feedback, we have made several changes to the paper that should help clarify many of the experimental details.
>
> >Other algorithms, such as off-policy algorithms (SAC, TD3)
>
> This is a good point! We would like to note that while SAC does display higher sample efficiency than PPO in the Mujoco gym environments, there are a few reasons why PPO is still arguably more popular. Off-policy algorithms tend to display much poorer wall-clock time and memory usage because of the replay buffer. Furthermore, PPO displays fairly similar performance in our investigated settings.
>
> The meta-optimisation process would thus take significantly longer, and we would rapidly run out of GPU memory because of the replay buffer. However, we are currently running experiments using DQN to investigate if the attack works on off-policy algorithms.
>
> >What are the useless features of the learning environments?
>
> In this paper, the useless features are the features that we append over the cheap talk channel. In general, useless features are features that do not influence the environment dynamics or reward. These appear in many real-world scenarios mentioned in the introduction and in currently-used RL environments. For example, [1] showed that two of the four inputs in the simple CartPole environment are useless.
>
> >constraining the length and values of the message
>
> Apologies for not making this clear! We constrain the message’s length and range of value -- we specify the length of the message in each environment in the hyperparameters in Appendix E. The value is between the range ($-2 \pi$, $2 \pi$). However, our PPO algorithm uses observation normalisation, so the range of values appearing in the appended messages are normalised and are not particularly important. We updated the paper to make this clearer.
>
> >how are the gradient updates shown on the x-axis and y-axis different?
>
> The gradient updates on the x-axis and y-axis in Figure 3 are the same. The plot shows the cosine distance between the different pairings. The differences between the gradient updates are described in Section 6.1. In detail, the gradient updates correspond to different buckets of timesteps within the environment. We reorganised the paper so that this description is now attached to the Figure.
>
> >the number of updates shown on the x-axis for RARL and ACT
>
> Apologies for the lack of clarity! We are showing the number of online RARL update phases on the X-Axis. At each point on the x-axis, we allow RARL to train against the Victim for 32k frames. Then, we train the Victim for 32k frames. This occurs at each “update” on the X-Axis. ACT is not allowed to update at all and is thus a horizontal line. We have since included the hyperparameters for this setting and clarified this in the updated paper.
>
> >Does proposition 1 hold for any tabular RL algorithm?
>
> Yes! We have updated the paper to make this clearer.
>
> >Which ψ is used in Figure 5?
>
> We used the final evolved $\psi$ at the end of meta-training.
>
> >The number of timesteps in Fig 1 and 5
>
> We selected the number of timesteps so the agent will have converged in training for each environment. The reason we do this is that ACT is highly compute-intensive. The environments vary significantly in difficulty and thus require a different number of timesteps to converge. This information is currently in Appendix E.
>
> >The runtime of training and test time attacks
>
> The longest attack (test-time Reacher) took 20 hours to train on 4 V100’s for 1024 generations. We updated the paper to include this information!
>
> >Is training large number of agents for different ES iterations realistic?
>
> This is a good question! One can imagine this attack would be hard to implement if training is expensive. However, note that adversarial attacks in machine learning are often more about investigating curious failure modes rather than suggesting practical attacks. It is quite surprising that we can cause RL algorithms to fail to learn a good policy by just appending features to the environment!
>
> >The authors should consider providing more details about how ES learns ϕ and ψ instead of treating it like black-box.
>
> Agreed! We have tried our best to directly visualise how $\phi$ and $\psi$ work in Figure 6 and in the videos attached to the supplementary. We included extra figures in the Appendix to provide more analysis and intuition.
>
> >The authors do not mention if they will make code available.
>
> We plan to make the code available! The code is a significant contribution of the work since the techniques described at the beginning of Section 6 are broadly useful.
>
> [1] Vischer, M. A., Lange, R. T., & Sprekeler, H. (2021). On lottery tickets and minimal task representations in deep reinforcement learning. arXiv preprint arXiv:2105.01648.

---

> > ### Comment · Reviewer_HHmW · 2022-11-18
> > **Post Response Comments**
> >
> > Dear Authors,
> >
> > Sorry for the delay, I was overwhelmed with work recently. Thanks for your response, it clarified a few things for me. However, I still do not see a general principle behind the identification of "useless features". You mentioned that [1] found this feature for CartPole, what about other environments? Is there a principled approach to finding them? This formulation is needed for studying attacks in more real-world environments.
> >
> > Another concern that still remains is the attack budget. The numerical range of [-2*\pi, 2*\pi] for the message range seems like the full range of the input feature. Further, it seems that the attacker is not limited by the number of steps to attack. Also, the attack requires access to large computational resources. These issues make the attack inefficient and while the attack may reveal some theoretical failure points, I am not sure if these vulnerabilities are practical and of interest to the ICLR audience.

---

> > > ### Author Response · Authors · 2022-11-19
> > > **Author Response to Comments**
> > >
> > > Thank you for taking the time to reply! We believe that [1] is a generally principled way of discovering useless features. However, note that our attack works on _any_ features. We _intentionally_ restrict it to useless ones to show that we are not influencing any relevant parts of the environment rewards or dynamics. In other words, the identification of useless features is not necessary for our attack to work; they are  just meant to show that our attack works _even if_ the features are useless. **In our paper we append features to the observation to guarantee that they are useless.** (We study this choice in Appendix H, where an Adversary that can control “useful” features trivially achieves far more powerful results).
> > >
> > > The notion of an “attack budget” (eg limiting numerical ranges and steps) is not something we consider. However, it is not necessarily the case that this would make the attack “inefficient” or any less practical than existing methods. In particular, most methods assume the ability to perform perturbations _across the entire observation_. This is arguably less realistic than having a large amount of control over a small number of features. (Note that we also included extra experimental results in that setting in Appendix H. This is still a novel setting since we are attacking a _learning_ agent rather than a static policy). Furthermore, we presented several plausibly practical settings in the introduction (eg attacking content recommendation systems by appending tags or attacking trading bots by placing orders far from the mid).
> > >
> > > We believe the results of the attack are relevant to the ICLR audience because they are counterintuitive and reveal a systematic way to generate shortcomings in current RL algorithms. It is likely the case that an algorithm that is robust to our attack would also perform better on existing environments and benchmarks since this type of catastrophic interference has previously been observed in existing Atari environments [2].
> > >
> > > [2] Fedus, W., Ghosh, D., Martin, J. D., Bellemare, M. G., Bengio, Y., & Larochelle, H. (2020). On catastrophic interference in atari 2600 games. arXiv.

---

> ### Author Response · Authors · 2022-11-18
> **Rebuttal Reminder**
>
> We would like to once again thank Reviewer HHmW for their review!
>
> We've made several updates to the manuscript to clarify many unclear parts of the paper that the reviewer pointed out! We've also just attached preliminary DQN results in the supplementary material. We fully expected the attack to also work in this setting since the Catastrophic Interference issues induced by ACT were discovered when analysing off-policy methods [1, 2]. Please let us know if we've addressed your concerns, and if not, which concerns still remain.
>
> The 18th of November marks the end of the first discussion phase, so we will no longer be able to update the submission after that point!
>
> [1] Fedus, W., Ghosh, D., Martin, J. D., Bellemare, M. G., Bengio, Y., & Larochelle, H. (2002). On catastrophic interference in atari 2600 games. arXiv.
>
> [2] Nikishin, E., Schwarzer, M., D’Oro, P., Bacon, P. L., & Courville, A. (2022, June). The primacy bias in deep reinforcement learning. In International Conference on Machine Learning (pp. 16828-16847). PMLR.

---

### Official Review · Reviewer_fBNt · 2022-11-02

**Confidence:** 3
**Correctness:** 3
**Technical Novelty And Significance:** 2
**Empirical Novelty And Significance:** 2
**Recommendation:** 6

**Clarity, Quality, Novelty And Reproducibility:**

- Novelty: This work presents a novel idea of appending adversarial attacks in the RL setting. The proposed setting is novel and well-motivated.

- Clarity: This work is well organized. However, the concepts and methods can be better explained and more details can be helpful for understanding the paper.
- Quality: This work has certain issues. These include my concerns about the technical contribution and experiment results, lack of baseline comparison etc.
- Reproducibility: I didn’t find the code of this paper. But I still trust the authors regarding their results.


**Strength And Weaknesses:**

#### **Strengths**:
- The appending attack and experiment settings are interesting
- The authors propose to leverage meta learning for generating attacks
- The proposed method is straightforward and demonstrates effectiveness

#### **Weakness**:
- The proposed Adversarial Cheap Talk (ACT) attack match the intuition of appending attack in the RL setting, but it seems the optimization is completely based on evolution strategies (ES), which makes the technical contribution of this work incremental.
- The two settings of train-time and test-time attacks seem to be less relevant. The former setting trains an adversary along with the victim and the latter setting uses backdoor for test-time manipulation.
- The test-time attacks introduce backdoor in training for test-time manipulation, however, it lacks details in section 5.3 of how the backdoor is designed and leveraged to fully control the victim.
- In the train-time influence experiment settings, the selected baseline methods (i.e., random and zero) are too naïve. For the RARL method, authors only present the results in the cartpole experiments, I would like to see more comparison in experiments like pendulum and reacher.
- In the test-time manipulation experiments (figure 5), why the performance of different settings is not consistent? For example, why does direct oracle reach the maximum return sooner in the reacher experiments?
- The adversary objective of test-time manipulation is to maximize the goal-conditioned scores, the setting of minimizing the return is not discussed.
- Finally, I am curious how the proposed attacks perform against simple defense methods as the authors discussed in the last section.


**Summary Of The Paper:**

This paper studies adversarial attacks in reinforcement learning under the setting where the adversary can only append to the agent’s observations. The authors propose a meta learning-based Adversarial Cheap Talk (ACT) method to generate such attacks, such that the agent’s training/test-time performance can be improved or attacked depending on the adversary’s objective. The authors also show that using ACT in training can affect test-time performance. Experiments demonstrate the effectiveness of the proposed ACT method.

**Summary Of The Review:**

Overall, I think this work is interesting. Moreover, the authors propose interesting attacks in the RL setting and have discussed train-time and test-time attacks. However, there are certain issues as I mentioned above, I think the paper would be in a better shape after taking the feedback into consideration.

---

> ### Author Response · Authors · 2022-11-14
> **Response to Reviewer fBNt**
>
> We thank Reviewer fBNt for their thorough review! It is great that the reviewer finds the paper to be “interesting”, “novel”, “straightforward”, and “effective”. We address the reviewer’s concerns below.
>
> # On the Test-Time Attacks
>
> Using the reviewer’s helpful feedback, we have made edits to this section to make it clearer! We summarise the clarifications below.
>
> >it lacks details in section 5.3 of how the backdoor is designed and leveraged to fully control the victim.
>
> We explain our meta-learning setup in Section 5.3. We attempt to interpret what the meta-learned backdoor does in Section 6.2 and Figure 6. In particular, Figures 6 (a) and (b) show that the meta-learned backdoor $\phi$ results in trained policies $\theta$ that have high correlation between the cheap talk channel and actions. Figure 6 (c) and (d) show that the $\theta$s consistently behave similarly across cheap talk channel values. If a consistent mapping from cheap talk channel inputs to output policy actions exists (and spans the space of actions), then $\psi$ should be able to learn a performant policy. We have also included more visualisations in the updated Appendix G.
>
> >The setting of minimizing the [test-time] return is not discussed.
>
> This is a good point! We updated the paper to include a brief discussion of this. In general, in the test-time setting, we can optimise for _any_ arbitrary objective, which includes the minimisation of test-time returns. We believe that minimising test-time returns is very easily achievable given that the current goal-conditioned test-time objectives we are achieving are far more difficult. We can include results on this setting if the reviewer finds it useful!
>
> >For example, why does direct oracle reach the maximum return sooner in the reacher experiments?
>
> This is a nice observation! There’s a good reason for this: The rate at which the adversaries learn in that setting seems to depend on how “aligned” the Victim’s and Adversary’s objectives are!
>
> In CartPole, the objectives are fairly aligned: Keeping the pole balanced helps reach the specific x-position. In particular, if the pole falls, the episode ends, and the agent cannot reach the target position. Thus, one can view the Victim’s training as a type of pre-training that helps with the “Oracle with $\phi$” scenario, while the “Direct Oracle” has to learn this behaviour from scratch.
>
> In Reacher, the objectives conflict with each other:  The Victim is trained to reach one position, but the adversary attempts to reach another arbitrary spot. Thus, one can view the “Oracle with $\phi$” scenario as having a type of adversarial pre-training, which makes it learn slower than the “Direct Oracle”.
>
> > The two settings of train-time and test-time attacks seem to be less relevant.
>
> We are uncertain precisely what the reviewer means by this. If the reviewer is saying that the settings seem unrelated, we would like to point the reviewer to our responses above, which should hopefully clarify the test-time attack setting!
>
> # General Comments
>
> > the optimization is completely based on evolution strategies (ES), which makes the technical contribution of this work incremental.
>
> Indeed, much of the contribution comes from the novel adversarial setting and its surprising results. However, note that there are significant technical contributions as well! In particular, we end-to-end JIT and vectorise the training of the RL algorithms for ES-based Meta-RL. This new technique allows us to achieve incredible training speeds (and is what enabled this project), allowing us to train >1 Million PPO Agents from scratch per hour on a single V100 GPU in CartPole.
>
> > the selected baseline methods (i.e., random and zero) are too naïve
>
> Since we introduced a new setting, there are few methods we can compare to. Indeed, “random” and “zero” are not baselines in the traditional sense -- these just represent standard RL training without the presence of the Ally or Adversary. We are unaware of other methods in the literature that we could directly compare to. If the reviewer could point us to some baselines they would like to see, that would be extremely useful!
>
> >how the proposed attacks perform against simple defense methods as the authors discussed
>
> In the last section, we discussed how addressing the train-time Adversary is quite challenging without domain knowledge. One can trivially “implement” this defence in our setting by just removing the Cheap Talk channels, in which case the Adversary cannot influence the Victim. For the test-time Adversary, we suggested a way to detect the test-time Adversary, but not a way to defend against it. If the reviewer has any ideas on simple defences, they would like us to try, that would be very useful!

---

> ### Author Response · Authors · 2022-11-17
> **Rebuttal Reminder**
>
> We would like to once again thank Reviewer fBNt fo their review! We did our best to take the reviewer's feedback into consideration to improve the paper and believe that it is now in better shape. We would like to kindly ask the reviewer to respond to let us know if we have addressed their concerns. The 18th of November marks the end of the first discussion phase, so we will no longer be able to update the submission after that point!

---

> > ### Comment · Reviewer_fBNt · 2022-11-17
> > **Response to the authors**
> >
> > Dear Authors, thank you very much for addressing my concerns. I will update my grade to support the acceptance of this work.

---

### Official Review · Reviewer_c6pu · 2022-11-03

**Confidence:** 4
**Correctness:** 3
**Technical Novelty And Significance:** 3
**Empirical Novelty And Significance:** 2
**Recommendation:** 6

**Clarity, Quality, Novelty And Reproducibility:**

Overall, the paper is well organized and clearly stated. There are a few places where improvements can be made.

1. In proposition 1, "the policy of a tabular Victim initialised uniformly along $\mathcal{M} is independent from its adversary$". Please use math in the proposition to give precise description of the quoted sentence. The description is embedded in the proof. But not every reader will read the proof and giving a rigorous and precise state is important.

2. In section 6.1, "Random Adversary: randomly initialise and fix the Adversary’s parameters". The description of random adversary is very vague. Adding "randomly" without specifying how the random initialization is done is misleading. Correct me if I missed any content.

I did not check the code. So I am not the right person to judge the reproducibility of the paper.

**Strength And Weaknesses:**

Strength:
1. The paper studies a novel and interesting problem, i.e., to craft adversarial attacks that require limited knowledge of the attacker and apply limited influence in the underlying MDP. And the crafted attacks can still achieve the adversary's objective.
2. The paper is nicely presented and easy to follow. Experiments are illustrative.


Weaknesses:

There are several places in the problem formulation that don't make sense, which needs further justification.

1. Previous work that studies adversarial attacks in the state observation (e.g., [R1]) models the attacks by $s'=s+\sigma$, where $s$ is the actual state, $s'$ is the observed state, and $\sigma$ is the attack crafted. Here, the ACT adversary append a message $m(s)$ to the actual state and assume that the victim use $(s,m(s))$ as the observation to generate its actions, which does not make too much sense from a security perspective. (a) Why the victim choose $(s,m(s))$ as the input of its strategy to choose its action? What is the motive?
(b). Of course, a RL agent does not know the model (the environment dynamics and the rewards). But the dimension of its state space is known beforehand. Appending a message is like changing the dimension of its state space, which would be easy to detect.

2. Does the assumption of stationarity benefit the attacker (to achieve its goal), the victim (to avoid non-convergence), or the authors (to show experimental results that can converge)? The authors did not justify the purpose of requiring the attacker not to introduce non-stationarity. What is the incentive for the attacker to do so If the attacker can achieve its goals by introducing non-stationarity to the learning process?

3. From the figure 1, when the adversary is in the "Ally" mode, the authors observe and state that the victim achieves higher rewards in training. But the authors did not explain why this happens? What is the root causes of it? How the attacker or the victim can leverage this phenomenon to do things?



[R1] Zhang, Huan, et al. "Robust reinforcement learning on state observations with learned optimal adversary." arXiv preprint arXiv:2101.08452 (2021).

**Summary Of The Paper:**

The paper studies adversarial attacks in RL. Unlike previous work, the paper considers adversarial attacks that append messages to the victim state observation. In the attack model, the attacker has very limited knowledge and the attacks don't intervene the the system dynamics, rewards, and remain deterministic during an episode. The contribution of the paper is proposing a new attack model called Adversarial Cheap Talk (ACT) that can achieve the attacker's goal with limited influence in the MDP and the victim.

**Summary Of The Review:**

The paper studies a novel and interesting problem. But there are caveats in the problem formulation that needs to be filled (Details are provided in section "Strength And Weaknesses").

---

> ### Author Response · Authors · 2022-11-14
> **Response to Reviewer c6pu**
>
> We thank Reviewer c6pu for their very thoughtful review! We are glad that the reviewer finds the paper “novel and interesting” and “easy to follow”. The reviewer brought up several well-reasoned conceptual questions about the problem formulation that we would like to address:
>
> >Appending a message is like changing the dimension of its state space, which would be easy to detect.
>
> This is a reasonable concern. However, the idea is not that a system designer would add a cheap-talk channel to their system but rather that such cheap-talk channels naturally occur in many real-world scenarios. We mention some real-world situations in the introduction (e.g. the recommender systems or financial models). However, practitioners currently also train with useless features in existing RL environments. Interestingly, [1] has found that even the simple CartPole environment only requires two of the four features to learn the optimal policy.
>
> We simulate these settings by appending to the observation; however, we could just as easily perturb the “useless” observations and expect similar results. For example, we are running experiments where we only perturb the useless features mentioned in [1].
>
> > Previous work that studies adversarial attacks in the state observation (e.g., [R1]) models the attacks by s′=s+σ
>
> This is a good point! We would like to point out a few things to help compare our attack vectors:
>
> 1. Our attack of appending can be seen as a strictly more limited version of the attack the reviewer mentioned (adjusting for perturbation-size constraints). One can view our setting as generating σ that can only perturb a subset of the features. It is arguably more realistic to consider an adversary that can only influence a subset of the observation rather than being able to apply arbitrary global perturbations.
>
> 2. We study the setting of perturbing useless features instead of perturbing all features because the act of perturbing all features can obscure relevant information in the state or conflate two different states. This can influence the global optimal performance in the underlying MDP (e.g. if it hides the ball in Pong or makes two different states appear the same). Meanwhile, if the adversary can only perturb useless features, it cannot influence the performance of the optimal policy (see our theoretical claims in Section 4).
>
> 3. We will run experiments using the attack vector the reviewer mentioned. This requires a trivial change to the code, and we can expect similar results for the reasons mentioned above. Note that this would still be a novel setting. While past works do use this attack vector, we are not aware of any works that meta-learn a stationary attack against a learning agent instead of attacking a static policy.
>
> We have updated the paper to clarify this comparison in Section 2.1 so that readers can better compare the works. We would also like to thank the reviewer for the reference -- it is highly relevant, and we have added it to the updated version of the paper.
>
> >On “the assumption of stationarity”
>
> This is a really interesting and subtle point! Initially, we had run experiments expecting that non-stationarity would be required for this attack to work. After all, without non-stationarity, we would merely be appending stationary and deterministic features to the agent’s observation (which the agent could easily learn to ignore).
>
> The current stationary adversary was intended to be an ablation to demonstrate this; however, to our surprise, we found that the adversary could still significantly influence performance (!!) We found that this subtle difference made for a far more interesting investigation that would be more broadly relevant to practitioners for understanding generic failure modes in RL.
>
> >What is the incentive for the attacker to be stationary?
>
> In real-world attacks, it would be far easier to implement a stationary adversary since it would just be a static function of the rest of the state. To implement a non-stationary attack, the adversary would need to know at what stage in training the victim is and at what point the training started. We have updated the paper to mention this!
>
> Furthermore, we think the stationary adversarial attack is more relevant for studying curious failure modes in RL in general. This is a common perspective taken in adversarial attacks in machine learning since most popular attacks are impractical.
>
> >On the Ally
>
> We briefly study how the Ally works in Appendix C Figure 7b. In general, we found these results to be less surprising. The Ally can, for example, learn to output the optimal policy logits or other useful information over the cheap talk channel.
>
> >On the clarity. fixes
>
> Thanks so much for pointing these out! We have fixed these in the updated version of the paper.
>
> [1] Vischer, M. A., Lange, R. T., & Sprekeler, H. (2021). On lottery tickets and minimal task representations in deep reinforcement learning. arXiv preprint arXiv:2105.01648.

---

> > ### Comment · Reviewer_c6pu · 2022-11-19
> > **Greatly appreciate the response**
> >
> > I really appreciate the replies from the authors. Please keep up the excellent work. These comments partially addressed my concerns. But I'd like to keep my original score. Below are some reasons. I hope the area chair can take these reasons into consideration when making a final call.
> >
> > "This is a reasonable concern. However, **the idea is not that a system designer would add a cheap-talk channel to their system but rather that such cheap-talk channels naturally occur in many real-world scenarios.** We mention some real-world situations in the introduction (e.g. the recommender systems or financial models). However, practitioners currently also train with useless features in existing RL environments. Interestingly, [1] has found that even the simple CartPole environment only requires two of the four features to learn the optimal policy."
> >
> > Again, I'd like to cast some doubts on the practicality of this attack model.
> >
> > First, practitioners train with useless features because practitioners don't know which features are useful. They leverage neural networks to extra useful features from a high-dimensional vector for decision-making. Attackers, falsifying the high-dimensional vector (e.g., an image), are unlikely to know which sub-vector (a set of pixels, the message m) is going to contribute to the useful features. Sometimes, this set of pixels carries important features, and sometimes they don't, especially in RL where we have a dynamic environment.
> >
> > Second, the real-world scenarios (e.g., the recommender systems and financial models) given in the paper don't make sense under this attacking model. First, how the adversary is going to add tags to the content? How tags does the adversary need to add to affect billions of data points? Second, even if the adversary can change the tags collectively, are these the messages that will not change the reward or dynamic of the environment? Third,  submitting orders far out of the money in the market or submitting many orders are hard to implement and easy to be detected. This is an article from the US department of justice. https://www.justice.gov/opa/pr/former-jp-morgan-traders-convicted-fraud-attempted-price-manipulation-and-spoofing-multi-year.
> >
> > Third, The authors mentioned, "simple CartPole environment only requires two of the four features". Please don't use CartPole as examples to show the practicality of the work. The CartPole environment is created for illustrative and educational purposes. In practice, it is easy to write the dynamic equation of the CartPole (since the physical law is not hard to write down, hence the model is known) and have a linear controller to stabilize it. There are two variables(features) that are needed the cart position and the pole angle. To derive a linear dynamic model, people also consider their speeds, which can be derived from the position and the angle. Using RL (taking images as input or taking the position and angle) to stabilize the cart and upright the pole is to create a simple example for people to play with. Hence, the CartPole environment cannot be used as an example to demonstrate practicality.
> >
> > "We study the setting of perturbing useless features instead of perturbing all features because the act of perturbing all features can obscure relevant information in the state or conflate two different states. "
> >
> > What I intended to ask is if the adversary wants to make the performance worse, why doesn't the adversary perturb the useful features as well, which seems to be more powerful in deteriorating the performance? If the adversary is introduced by the practitioner to make the performance more robust, we should clearly state how it can improve the robustness.

---

> > > ### Author Response · Authors · 2022-11-19
> > > **Author Response to Comment**
> > >
> > > We would like to thank the reviewer for their reply and their supportive comments! The reviewer has brought up some reasonable points we’d like to address.
> > >
> > > # On useful features
> > >
> > > We think there is one common point could address a few of these concerns: The reviewer is correct in that, for this attack to work, the features don’t have to be useless! In fact the attack works far better if the features are useful (see Appendix H). This is an unsurprising result since the attacker can obscure relevant information.
> > >
> > > We show that the attack works _even if_ the features aren’t useful. This is counterintuitive and is why we find the results interesting. In theory, the Victim should just learn to ignore the useless features. We intentionally limited the range of influence of the adversary to show that, in the worst case scenario where the adversary can only affect irrelevant information (like background information in an image), the attack surprisingly still works.
> > >
> > > > practitioners train with useless features because practitioners don't know which features are useful.
> > >
> > > We completely agree! This is _why_ we believe our setting is reasonable.
> > >
> > > > Sometimes, this set of pixels carries important features, and sometimes they don't.
> > >
> > > We agree again! The point is that in the _worst-case_ scenario, where the pixels aren’t important, the adversary can still influence performance. It is also reasonable to imagine scenarios where this would be the case (Eg if the attacker can only influence pixels by arranging objects in the background of the image).
> > >
> > > To summarize this point: our results show that the attacker can still strongly influence a victim’s performance in the worst-case scenario that it can only influence useless features. The _same exact attack_ can be done on useful features (and would still be a novel setting), but we intentionally _limit_ the adversary (akin to an attack budget in past work) to show its effectiveness.
> > >
> > > # On the real-world scenarios
> > >
> > > The reviewer brings up many good points here!  It is difficult to discuss real-world versions of this attack because RL is not yet widely-used in the real-world. Beyond real-world scenarios, we think the attack is relevant for studying curious failure modes in RL in general. It is likely the case that an algorithm that is robust to our attack would also perform better on existing environments and benchmarks since the type of catastrophic interference induced by the adversary has previously been observed to naturally exist in Atari environments [2].
> > >
> > > [2] Fedus, W., Ghosh, D., Martin, J. D., Bellemare, M. G., Bengio, Y., & Larochelle, H. (2020). On catastrophic interference in atari 2600 games. arXiv.
> > >
> > > > On appending tags to the content
> > >
> > > In this scenario, one can imagine an entity (eg a group of creators) that produces a significant proportion of the content on a platform. This entity would usually be able to append tags to its content for SEO. These tags wouldn’t influence user behavior (which would be the environment dynamics) since they wouldn’t be visible to the user; however, they could influence the platform’s models.
> > >
> > > > On out-of-the-money orders
> > >
> > > We are just suggesting that this is an attack that could hypothetically be done. We are not suggesting that it would be legal, undetectable, or easy to implement.
> > >
> > > # On Cartpole
> > >
> > > We only mentioned CartPole to show that it is very common for people to train with useless features. We weren’t trying to say anything more than that. The same paper [1] showed that many features in the Minatar environments are also useless. One can imagine that the background in procgen is also useless. While these settings aren’t “practical”, very few (if any) of the existing RL settings commonly studied by researchers are practical.

---

> ### Author Response · Authors · 2022-11-18
> **Rebuttal Period Reminder**
>
> Thanks again for taking the time to write such a thoughtful review! We hope we addressed the reviewer's concerns in our response, updated manuscripts, and new experiments. Please let us know if you still have any further questions or concerns after reading our response!
>
> The 18th of November marks the end of the first discussion phase, so we will no longer be able to update the submission after that point!

---

### Author Response · Authors · 2022-11-17
**General Response**

We thank the reviewers for their thorough, thoughtful, and helpful feedback! Overall, we are glad that reviewers generally found the paper to be novel, interesting, and straightforward. We have used the reviewers' feedback to update the writing in the paper to be clearer and added extra experiments.

In Appendix G we added more visualisations of $\theta$ when trained with a learned $\phi$ or a random $\phi$.

In Appendix H, we added results where the agent adds a perturbation to the observation instead of modifying a cheap talk channel. The adversary here is much more powerful since it can actual obscure information and conflate observations. Thus, it can much more severely harm performance.

Also in Appendix H, we added results where the agent can only perturb the “useless” inputs identified in [1]. As expected, we get similar results to when we use the cheap talk channel.

[1] Vischer, M. A., Lange, R. T., & Sprekeler, H. (2021). On lottery tickets and minimal task representations in deep reinforcement learning. arXiv preprint arXiv:2105.01648.

---

### Decision · Program_Chairs · 2023-01-20

**Decision:**

Reject

**Comment:**

Additional references from AC:

Zhang, Huan, Hongge Chen, Chaowei Xiao, Bo Li, Mingyan Liu, Duane Boning, and Cho-Jui Hsieh. "Robust deep reinforcement learning against adversarial perturbations on state observations." Advances in Neural Information Processing Systems 33 (2020): 21024-21037.

Sun, Yanchao, Ruijie Zheng, Yongyuan Liang, and Furong Huang. "Who Is the Strongest Enemy? Towards Optimal and Efficient Evasion Attacks in Deep RL." In International Conference on Learning Representations.

Shen, Qianli, Yan Li, Haoming Jiang, Zhaoran Wang, and Tuo Zhao. "Deep reinforcement learning with robust and smooth policy." In International Conference on Machine Learning, pp. 8707-8718. PMLR, 2020.

Oikarinen, Tuomas, Wang Zhang, Alexandre Megretski, Luca Daniel, and Tsui-Wei Weng. "Robust deep reinforcement learning through adversarial loss." Advances in Neural Information Processing Systems 34 (2021): 26156-26167.

Korkmaz, Ezgi. "Investigating vulnerabilities of deep neural policies." In Uncertainty in Artificial Intelligence, pp. 1661-1670. PMLR, 2021.

Behzadan, Vahid, and Arslan Munir. "Whatever does not kill deep reinforcement learning, makes it stronger." arXiv preprint arXiv:1712.09344 (2017).

Mandlekar, Ajay, Yuke Zhu, Animesh Garg, Li Fei-Fei, and Silvio Savarese. "Adversarially robust policy learning: Active construction of physically-plausible perturbations." In 2017 IEEE/RSJ International Conference on Intelligent Robots and Systems (IROS), pp. 3932-3939. IEEE, 2017.

Pattanaik, Anay, Zhenyi Tang, Shuijing Liu, Gautham Bommannan, and Girish Chowdhary. "Robust Deep Reinforcement Learning with Adversarial Attacks." In Proceedings of the 17th International Conference on Autonomous Agents and MultiAgent Systems, pp. 2040-2042. 2018.

Lütjens, Björn, Michael Everett, and Jonathan P. How. "Certified adversarial robustness for deep reinforcement learning." In Conference on Robot Learning, pp. 1328-1337. PMLR, 2020.

Fischer, Marc, Matthew Mirman, Steven Stalder, and Martin Vechev. "Online robustness training for deep reinforcement learning." arXiv preprint arXiv:1911.00887 (2019).

Kumar, Aounon, Alexander Levine, and Soheil Feizi. "Policy Smoothing for Provably Robust Reinforcement Learning." In International Conference on Learning Representations.

Wu, Fan, Linyi Li, Zijian Huang, Yevgeniy Vorobeychik, Ding Zhao, and Bo Li. "CROP: Certifying Robust Policies for Reinforcement Learning through Functional Smoothing." In International Conference on Learning Representations.

Pan, Xinlei, Chaowei Xiao, Warren He, Shuang Yang, Jian Peng, Mingjie Sun, Mingyan Liu, Bo Li, and Dawn Song. "Characterizing Attacks on Deep Reinforcement Learning." In Proceedings of the 21st International Conference on Autonomous Agents and Multiagent Systems, pp. 1010-1018. 2022.

Tan, Kai Liang, Yasaman Esfandiari, Xian Yeow Lee, and Soumik Sarkar. "Robustifying reinforcement learning agents via action space adversarial training." In 2020 American control conference (ACC), pp. 3959-3964. IEEE, 2020.

Tessler, Chen, Yonathan Efroni, and Shie Mannor. "Action robust reinforcement learning and applications in continuous control." In International Conference on Machine Learning, pp. 6215-6224. PMLR, 2019.

Lee, Xian Yeow, Yasaman Esfandiari, Kai Liang Tan, and Soumik Sarkar. "Query-based targeted action-space adversarial policies on deep reinforcement learning agents." In Proceedings of the ACM/IEEE 12th International Conference on Cyber-Physical Systems, pp. 87-97. 2021.

Behzadan, Vahid, and Arslan Munir. "Vulnerability of deep reinforcement learning to policy induction attacks." In Machine Learning and Data Mining in Pattern Recognition: 13th International Conference, MLDM 2017, New York, NY, USA, July 15-20, 2017, Proceedings 13, pp. 262-275. Springer International Publishing, 2017.

Huang, Yunhan, and Quanyan Zhu. "Deceptive reinforcement learning under adversarial manipulations on cost signals." In Decision and Game Theory for Security: 10th International Conference, GameSec 2019, Stockholm, Sweden, October 30–November 1, 2019, Proceedings 10, pp. 217-237. Springer International Publishing, 2019.

Sun, Yanchao, Da Huo, and Furong Huang. "Vulnerability-Aware Poisoning Mechanism for Online RL with Unknown Dynamics." In International Conference on Learning Representations.

Zhang, Xuezhou, Yuzhe Ma, Adish Singla, and Xiaojin Zhu. "Adaptive reward-poisoning attacks against reinforcement learning." In International Conference on Machine Learning, pp. 11225-11234. PMLR, 2020.

Rakhsha, Amin, Goran Radanovic, Rati Devidze, Xiaojin Zhu, and Adish Singla. "Policy teaching via environment poisoning: Training-time adversarial attacks against reinforcement learning." In International Conference on Machine Learning, pp. 7974-7984. PMLR, 2020.


**Justification For Why Not Higher Score:**

The paper needs major clarifications to (1) justify more on why this setting is practical, and (2) distinguish how this is different from state perturbations in a blackbox manner. In particular, (2) is rather important as the concept of useless features, which are defined as the features that are do not affect the transition and reward, are not necessarily identifiable to attackers. If attackers cannot identify useless features, how does this attack model differ from a black-box state perturbation? Overall, when proposing a rather unconventional setting, it is beneficial to provide sufficient discussions on how this proposed setting positions in the existing literature.

In addition, the paper lacks some references to very recent advances in adversarial/robust RL.

Finally, there is no discussion about optimality or sub-optimality of the proposed attack.

**Justification For Why Not Lower Score:**

N/A

**Metareview: Summary, Strengths And Weaknesses:**

The paper studies adversarial attacks in RL, but considers an unconventional setting that the adversarial attacks append messages to the victim's state observation. Using this new setting of Cheap Talk MDPs, the authors proved that this only allows for a minimal range of influence. Specifically, they show that adversaries cannot influence Victims in the tabular setting, irrespective of the Victim’s learning algorithm. Such adversaries can only attack Victims by interfering with their function approximator for deep RL.

The reviewers have a lot of concerns about justifications of the setting. The paper needs major clarifications to (1) justify more on why this setting is practical, and (2) distinguish how this is different from state perturbations in a blackbox manner. In particular, (2) is rather important as the concept of useless features, which are defined as the features that do not affect the transition and reward, are not necessarily identifiable to attackers. If attackers cannot identify useless features, how does this attack model differ from a black-box state perturbation? Overall, when proposing a rather unconventional setting, it is beneficial to provide sufficient discussions on how this proposed setting positions in the existing literature.

In addition, the paper lacks some references to very recent advances in adversarial/robust RL (see comment).

Finally, there is no discussion about optimality or sub-optimality of the proposed attack.

Overall, the paper introduces an interesting and novel setting. The authors are encouraged to provide more convincing justifications and a more thorough comparison with the existing literature to make the paper even stronger.